# Learning Mixtures of Linear Dynamical Systems via Hybrid Tensor-EM Method

**Lulu Gong and Shreya Saxena**
Department of Biomedical Engineering
Wu Tsai Institute, Center for Neurocomputation and Machine Intelligence
Yale University, New Haven, CT, 06510, USA
{lulu.gong}@yale.edu

## Abstract

Linear dynamical systems (LDSs) have been powerful tools for modeling high-dimensional time-series data across many domains, including neuroscience. However, a single LDS often struggles to capture the heterogeneity of neural data, where trajectories recorded under different conditions can have variations in their dynamics. Mixtures of linear dynamical systems (MoLDS) provide a path to model these variations in temporal dynamics for different observed trajectories. However, MoLDS remains difficult to apply in complex and noisy settings, limiting its practical use in neural data analysis. Tensor-based moment methods can provide global identifiability guarantees for MoLDS, but their practical performance degrades in high-noise or complex scenarios. Commonly used expectation-maximization (EM) methods offer flexibility in fitting latent models but are highly sensitive to initialization and prone to poor local minima. Here, we propose a tensor-based moment method that provides identifiability guarantees for learning MoLDS, which can be followed by EM updates to combine the strengths of both approaches. The novelty in our approach lies in the construction of moment tensors using the input-output data, on which we then apply Simultaneous Matrix Diagonalization (SMD) to recover globally consistent estimates of mixture weights and system parameters. These estimates can then be refined through a full Kalman EM algorithm, with closed-form updates for all LDS parameters. We validate our framework on synthetic benchmarks and real-world datasets. On synthetic data, the proposed Tensor-EM method achieves more reliable recovery and improved robustness compared to either pure tensor or randomly initialized EM methods. We then apply this method to two neural datasets from non-human primates doing reaching tasks. For both datasets, our method successfully models and clusters different conditions as separate subsystems. These results demonstrate that MoLDS provides an effective framework for modeling complex neural data in different brain regions, and that Tensor-EM is a principled and reliable approach to MoLDS learning for these applications.

## 1 Introduction

Neuroscience experiments now produce large volumes of high-dimensional time-series datasets, calling for new computational tools to uncover the dynamical principles underlying brain function (Paninski & Cunningham, 2018; Stringer & Pachitariu, 2024; Urai et al., 2022; Saxena & Cunningham, 2019). These recorded data often originate from multiple, distinct underlying dynamical processes, yet the identity of the generating systems at any given time is unknown. Estimating and recovering the parameters and dynamics of such latent systems from these mixed trajectories is a central challenge in system identification and machine learning (Ljung, 1998; Durbin & Koopman, 2012; Bishop, 2006).

Classical mixture models, such as mixtures of Gaussians (MoG) (Dempster et al., 1977) and mixtures of linear regressions (MLR) (De Veaux, 1989), provide valuable tools for modeling these heterogeneous data. However, they are primarily designed for static settings and do not explicitly capture temporal dependencies, limiting their applicability to sequential data where temporal dynamics are central. In contrast, dynamical models such as linear dynamical systems (LDS) and their modern

extensions, such as the (recurrent) switching LDS (SLDS) and the decomposed LDS (dLDS) (Ghahramani & Hinton, 2000; Fox, 2009; Linderman et al., 2017; Mudrik et al., 2024; Zhang & Saxena, 2024), are suitable for time-series data modeling with latent states and potential regime switches. Switching models typically target a single long trajectory and are prone to solutions that contain frequent switching; they are not suitable when the goal is to learn parameters of multiple distinct LDSs from independent trials, which is precisely the common structure in neural experiments.

Mixtures of linear dynamical systems (MoLDS) (Chen & Poor, 2022; Bakshi et al., 2023; Rui & Dahleh, 2025) effectively address this setting by treating each trajectory as originating from a single latent LDS. Then the learning of MoLDS aims to uncover the number of subsystems, their parameters, and the mixture weights from collections of input-output trajectories. This formulation enables direct parameter recovery and interpretability, making it appealing for many applications, including large-scale neural data analysis. Neural datasets usually have large numbers of recordings across behavioral or task conditions, with many trials under each condition. An important question is whether neural populations reuse common latent dynamics across conditions or exhibit distinct dynamics for different behaviors (Vyas et al., 2020; Athalye et al., 2023). MoLDS can provide a principled way to explore this question: by identifying shared and condition-specific latent dynamics, it can reveal the mixture structure of all trials in the dataset and offer a comprehensive view of the underlying dynamical structures and motifs.

For the inference of MoLDS, tensor-based moment methods are commonly employed, where high-order statistical moments of the input-output data are reorganized into structured tensors, and their decomposition provides estimates of mixture weights and LDS parameters. The appeal of these algebraic approaches lies in their global identifiability: unlike iterative optimization methods that navigate non-convex landscapes, tensor decomposition exploits the algebraic structure of moments to directly recover parameters through polynomial equation systems that admit unique solutions under ideal conditions (Anandkumar et al., 2014). However, their practical performance is often limited because moment estimates become imperfect in realistic, noisy datasets, leading to degraded parameter recovery (Kuleshov et al., 2015). In parallel, likelihood-based approaches such as expectation-maximization (EM) have long been widely used for fitting classical mixture models and LDSs. In this context, EM provides a flexible iterative procedure for jointly estimating latent states, mixture weights, and system parameters through local likelihood optimization. While powerful and widely adopted, EM suffers from well-known sensitivity to initialization and susceptibility to poor local minima (Xu & Jordan, 1996; Bishop, 2006). These limitations become particularly problematic in the MoLDS setting where the parameter space is high-dimensional and the likelihood surface is highly multimodal.

Here, we propose a hybrid Tensor-EM framework that strategically combines global initialization with local refinement of these two methods. We first apply tensor decomposition based on Simultaneous Matrix Diagonalization (SMD) (Kuleshov et al., 2015) to obtain stable and accurate initial estimates. We then use these estimates to initialize a full Kalman filter-smoother EM procedure, which refines all parameters through closed-form updates over all trajectories. This hybrid approach harnesses the global identifiability of tensor methods for robust initialization, while leveraging EM's superior local optimization, to achieve both reliability and efficiency.

We validate this framework on both synthetic and real-world neural datasets. On synthetic benchmarks, the proposed Tensor-EM method achieves more reliable recovery and improved robustness compared to (i) pure tensor methods and (ii) EM with random initialization. Next, we analyze neural recordings from two different experiments: (1) Recordings from monkey somatosensory cortex during center-out reaches in different directions, where Tensor-EM identifies distinct dynamical clusters corresponding to the reaching directions, matching supervised LDS fits per direction but achieved in a fully unsupervised manner; (2) Recordings from the dorsal premotor cortex while a monkey performs sequential reaches in more distributed directions, where Tensor-EM succeeds in parsing the different trials into direction-specific dynamical models. These results establish MoLDS as an effective framework for modeling heterogeneous neural systems, and demonstrate that Tensor-EM provides a principled and reliable solution for learning MoLDS in both synthetic and challenging real-world settings.

## 2   RELATED WORK

**Mixtures models.** Classical mixture models (e.g., MoG and MLR) capture heterogeneity but not explicit temporal structure (Dempster et al., 1977; De Veaux, 1989; Li & Liang, 2018). Importantly, MoLDS is related to MLR through lagged-input representations: by augmenting inputs with their past values, an MLR model can approximate certain temporal dependencies (Rui & Dahleh, 2025). Through this connection, MoLDS inherits useful algorithmic tools, including spectral tensor methods and optimization approaches (Anandkumar et al., 2014; Yi et al., 2016; Pal et al., 2022; Li & Liang, 2018), while maintaining its superior modeling capacity for dynamical systems.

**LDS models and their variants.** LDS models have been widely used to model time-series data, and several extensions have been developed to better handle nonstationarity and nonlinear structures. SLDS methods (Ghahramani & Hinton, 2000; Linderman et al., 2017) model long trajectories that switch between different dynamical regimes over time, requiring joint inference of both continuous latent states and discrete mode sequences. The dLDS method (Mudrik et al., 2024; Chen et al., 2024) captures more gradual or overlapping changes by expressing dynamics as sparse, time-varying combinations of basis operators. These frameworks are designed for long, nonstationary sequences where the dynamics themselves evolve over time. In contrast, the MoLDS setting assumes that each short trajectory is well described by a single LDS drawn from a collection of LDS components. The goal then is to identify the set of latent dynamical systems and assign trials to those components, rather than modeling intra-trial dynamics changes.

**Tensor methods and EM.** Tensor decomposition methods offer a principled algebraic approach to parameter estimation in latent variable models, with polynomial-time algorithms and theoretical identifiability guarantees (Anandkumar et al., 2014; Panagakis et al., 2021). Recent work in tensor-based MoLDS learning has explored different moment construction strategies and decomposition algorithms. Early approaches applied Jennrich's algorithm directly to input-output moments (Bakshi et al., 2023), while more recent work incorporates temporal lag structure using the MLR reformulation and the robust tensor power method (Rui & Dahleh, 2025). Our approach adopts the Simultaneous Matrix Diagonalization (SMD) method (Kuleshov et al., 2015), which also operates on whitened tensor slices and offers improved numerical stability and robustness to noise, which are critical advantages in the challenging MoLDS setting and neural data.

The combination of tensor initialization followed by EM-style refinement has been proven effective in mixture model settings. In the MLR literature, tensor methods have been shown to provide globally consistent initial estimates that lie within the basin of attraction of the maximum likelihood estimator, which are then refined using alternating minimization (Yi et al., 2016; Zhong et al., 2016; Chen et al., 2021). However, alternating minimization represents a simplified version of EM that uses hard assignments rather than the probabilistic responsibilities essential for handling uncertainty in noisy settings. Our work extends this paradigm to the more complex MoLDS setting by combining tensor initialization with more powerful Kalman filter-smoother EM, including proper handling of latent state inference and closed-form parameter updates.

**Contributions.** Our work makes both methodological and empirical contributions to MoLDS learning and shows its practical utility in neuroscience settings. Methodologically, relative to existing MoLDS tensor methods, our approach makes several key advances: (i) we employ SMD for more stable decomposition of whitened moment tensors, (ii) we provide a principled initialization strategy for noise parameters $(Q, R)$ based on residual covariances, which were missing from prior tensor-based approaches, and (iii) we integrate a complete EM procedure with responsibility-weighted sufficient statistics for all LDS parameter updates. Compared to existing tensor-alternating minimization pipelines for MLR, our method leverages the full flexibility of Kalman filtering and smoothing, which is essential for addressing the temporal dependencies and uncertainty quantification required in real MoLDS applications.

Empirically, we demonstrate the successful applications of Tensor-EM MoLDS methods to complex and real-world data analysis. While prior MoLDS tensor work has been limited to synthetic evaluations, we show that our Tensor-EM framework can effectively analyze neural recordings from different brain regions during distinct reaching tasks and successfully identify distinct dynamical regimes corresponding to different movement directions in a fully unsupervised manner. This represents an important step toward making MoLDS a practical tool for important applications, particularly in neuroscience, where capturing heterogeneous dynamics across experimental conditions is a central

challenge. Together, these methodological and empirical advances demonstrate that our Tensor-EM MoLDS framework provides improved robustness and accuracy in both controlled and challenging real-world settings.

# 3 MoLDS: Model and Tensor-EM Method

## 3.1 Mixture of Linear Dynamical Systems (MoLDS)

In the MoLDS setting (Figure 1), we observe $N$ input-output trajectories $\{(u_{i,0:T_i-1}, y_{i,0:T_i-1})\}_{i=1}^{N}$, each generated by one of $K$ latent LDS components. Let $z_i \in [K]$ denote the (unknown) component for trajectory $i$, drawn i.i.d. as $z_i \sim \text{Multinomial}(p_{1:K})$, where $p_k \in (0,1)$ are the mixture weights with $\sum_{k=1}^{K} p_k = 1$, indicating the probability of a trajectory being generated by component $k$. Conditional on $z_i = k$, the data is generated from the following LDS:

$$x_{t+1} = A_k x_t + B_k u_t + w_t, \qquad w_t \sim \mathcal{N}(0, Q_k), \tag{1}$$

$$y_{t+1} = C_k x_t + D_k u_t + v_t, \qquad v_t \sim \mathcal{N}(0, R_k), \tag{2}$$

with $A_k \in \mathbb{R}^{n \times n}$, $B_k \in \mathbb{R}^{n \times m}$, $C_k \in \mathbb{R}^{p \times n}$, $D_k \in \mathbb{R}^{p \times m}$, and $Q_k \succeq 0$, $R_k \succeq 0$. The goal of MoLDS learning is to recover the mixture weights and LDS parameters $\{p_k, (A_k, B_k, C_k, D_k, Q_k, R_k)\}_{k=1}^{K}$. These parameters are identifiable only up to two natural ambiguities: the ordering of the components (permutation) and similarity transformations of the latent state realization (which leave the input-output behavior unchanged). [1]

To make this recovery possible, we adopt several standard conditions: (i) inputs are persistently exciting, (ii) each LDS component is controllable and observable, and (iii) the components are sufficiently separated to ensure identifiability (Bakshi et al., 2023; Rui & Dahleh, 2025).

## 3.2 Tensor-EM Approach Overview

Our complete approach (Algorithm 1) consists of two stages: a tensor initialization stage (see Algorithm 3 in the Appendix), which provides globally consistent estimates of the mixture weights and system parameters, and an EM refinement stage (see Algorithm 4), which further improves these estimates and achieves statistical efficiency. Between these two stages, a key step is the initialization of the noise parameters $(Q_k, R_k)$, since these are not identifiable from the tensor-based estimates. We address this gap by estimating them from the residual covariances computed using the tensor-based parameter estimates (detailed in Appendix D).

This Tensor-EM approach combines the global identifiability guarantees of algebraic methods with the statistical optimality of likelihood-based inference. This hybrid approach is particularly effective in challenging settings with limited data, high noise, or poor component separation – scenarios where neither pure tensor methods nor randomly initialized EM perform reliably.

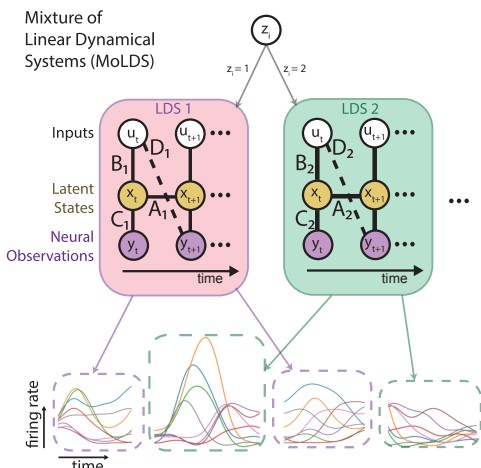

Figure 1: Overview of MoLDS and the application to neural data analysis.

## 3.3 Tensor Initialization for MoLDS

The tensor initialization leverages the key insight that MoLDS can be reformulated as MLR through lagged input representations (Rui & Dahleh, 2025). This transformation exposes the mixture structure in high-order moments, enabling algebraic recovery of component parameters via tensor decomposition (see Appendix B for details). The method works by exploiting the impulse-response (Markov

---

[1]Formally, any invertible matrix $M$ inducing an equivalent realization via $A_k \rightarrow M^{-1}A_k M$, $B_k \rightarrow M^{-1}B_k$, $C_k \rightarrow C_k M$, and $D_k$ unchanged, will yield the same input-output mapping.

---

**Algorithm 1** Tensor-EM Pipeline for MoLDS

---

**Require:** Trajectories $\{(u_{i,0:T_i-1}, y_{i,0:T_i-1})\}_{i=1}^N$, truncation length $L$, LDS order $n$, #components $K$

**Ensure:** Mixture weights $\{\hat{p}_k\}_{k=1}^K$ and LDS parameters $\{(\hat{A}_k, \hat{B}_k, \hat{C}_k, \hat{D}_k, \hat{Q}_k, \hat{R}_k)\}_{k=1}^K$

    **Stage 1: Tensor Initialization**                 (Appendix Alg. 3)

 1: Transform MoLDS to MLR via lagged inputs; construct moment tensors $M_2$, $M_3$

 2: Apply whitening and SMD to recover mixture weights $\{\hat{p}_k\}$ and Markov parameters

 3: Realize LDS matrices $\{(\hat{A}_k, \hat{B}_k, \hat{C}_k, \hat{D}_k)\}$ via Ho-Kalman algorithm

 4: Initialize noise parameters $\{(\hat{Q}_k^{(0)}, \hat{R}_k^{(0)})\}$ from residual covariances       (Appendix D)

    **Stage 2: EM Refinement**                      (Appendix Alg. 4)

 5: **repeat**

 6:     **E-step:** Compute trajectory responsibilities $\gamma_{i,k}$ via Kalman filter likelihoods

 7:     Run Kalman smoother to obtain responsibility-weighted sufficient statistics

 8:     **M-step:** Update mixture weights and all LDS parameters via closed-form MLE

 9: **until** convergence in log-likelihood

10: **return** Refined parameters $\{\hat{p}_k, (\hat{A}_k, \hat{B}_k, \hat{C}_k, \hat{D}_k, \hat{Q}_k, \hat{R}_k)\}_{k=1}^K$

---

parameter) representation of LDSs. We first construct second- and third-order cross moments of the lagged input-output data, denoted as $M_2$ and $M_3$

$$M_2 \;=\; \frac{1}{2|\mathcal{N}_2|} \sum_{j \in \mathcal{N}_2} \tilde{y}_j^2 \left(v_j \otimes v_j - I_d\right), \qquad M_3 \;=\; \frac{1}{6|\mathcal{N}_3|} \sum_{j \in \mathcal{N}_3} \tilde{y}_j^3 \left(v_j^{\otimes 3} - \mathcal{E}(v_j)\right), \qquad (3)$$

where $\mathcal{N}_2$ and $\mathcal{N}_3$ are two disjoint subsets of samples, $\tilde{y}_j = y_{i,t}$ is the observed output, $v_j$ is the normalized lagged input, $I_d$ and $\mathcal{E}$ are correction terms (see Appendix B.2 for notations). Then we apply the whitening transformation $W$ and decompose the resulting tensor into weights and Markov parameter estimates. At last, we recover LDS state-space parameters through the Ho-Kalman realization (Oymak & Ozay, 2019). The core mathematical insight is that if $M_2$ and $M_3$ are appropriately constructed from sub-sampled lagged covariates, the whitened tensor

$$\widehat{T} \;=\; M_3(W, W, W) \;=\; \sum_{k=1}^K p_k \, \alpha_k^{\otimes 3} \qquad (4)$$

is symmetric and orthogonally decomposable, where $\alpha_k \in \mathbb{R}^K$ are the whitened representations of the regression vectors (scaled Markov parameters) for each LDS component, and $p_k$ are the corresponding mixture weights. This decomposition is unique and recovers the mixture components $\{\alpha_k, p_k\}$ up to permutation and sign. Importantly, we employ SMD (Kuleshov et al., 2015) for this decomposition step, which is shown to be more stable and accurate empirically in Section 4.1. We then apply the Ho-Kalman realization (Oymak & Ozay, 2019) to recover the state-space parameters for each component, which are then used as principled initializations for the subsequent refinement stage. The complete procedure is provided in Algorithm 3 in Appendix B.4, together with the MoLDS-to-MLR reformulation and tensor construction details in Appendix B.

## 3.4 EM Refinement for MoLDS

The tensor initialization provides globally consistent estimates of mixture weights and system matrices, but does not recover the noise parameters $(Q_k, R_k)$ nor achieve optimal statistical accuracy. We therefore need to refine these estimates using a full Kalman filter-smoother EM algorithm that maximizes the observed-data likelihood.

Our EM formulation extends classical mixture EM to the MoLDS setting by computing trajectory-wise responsibilities via Kalman filter likelihoods, then updating parameters from responsibility-weighted sufficient statistics (see Appendix C and Algorithm 4 for details). In brief, at iteration $t$, given current parameters $\hat{\theta}^{(t)} = \{(\hat{p}_k^{(t)}, \hat{A}_k^{(t)}, \hat{B}_k^{(t)}, \hat{C}_k^{(t)}, \hat{D}_k^{(t)}, \hat{Q}_k^{(t)}, \hat{R}_k^{(t)})\}_{k=1}^K$, the E-step computes responsibilities

$$\gamma_{i,k}^{(t)} \;=\; \frac{\exp\left(\log \hat{p}_k^{(t)} + \log p(y_{i,0:T_i-1} \mid u_{i,0:T_i-1}, \hat{\theta}_k^{(t)})\right)}{\sum_{k=1}^K \exp\left(\log \hat{p}_k^{(t)} + \log p(y_{i,0:T_i-1} \mid u_{i,0:T_i-1}, \hat{\theta}_k^{(t)})\right)}. \qquad (5)$$

Next, we use a Kalman smoother to compute responsibility-weighted sufficient statistics $S_k^{(t)}$ for each component. The M-step then updates all parameters via closed-form maximum likelihood estimates (MLE)

$$(\hat{A}_k^{(t+1)}, \hat{B}_k^{(t+1)}, \hat{C}_k^{(t+1)}, \hat{D}_k^{(t+1)}, \hat{Q}_k^{(t+1)}, \hat{R}_k^{(t+1)}) = \text{MLE-LDS}(S_k^{(t)}), \tag{6}$$

$$\hat{p}_k^{(t+1)} = \frac{1}{N} \sum_i \gamma_{i,k}^{(t)}, \quad \forall k \in [K]. \tag{7}$$

## 4 TENSOR-EM PERFORMANCE ON SYNTHETIC DATA

### 4.1 SMD-TENSOR METHOD PROVIDES MORE RELIABLE RECOVERY ON SIMULATED MOLDS

We demonstrate that the employed SMD-Tensor method reliably recovers parameters in synthetic MoLDS with a small number of components, low latent dimensionality, and well-separated dynamics. We empirically compare SMD-Tensor against the RTPM-Tensor baseline across a wide range of such settings. Our results show that SMD-Tensor consistently outperforms RTPM-Tensor on multiple metrics. In particular, when the number of mixture components is small and sufficient data are available, SMD achieves near-optimal recovery of both the mixture weights and the Markov parameters.

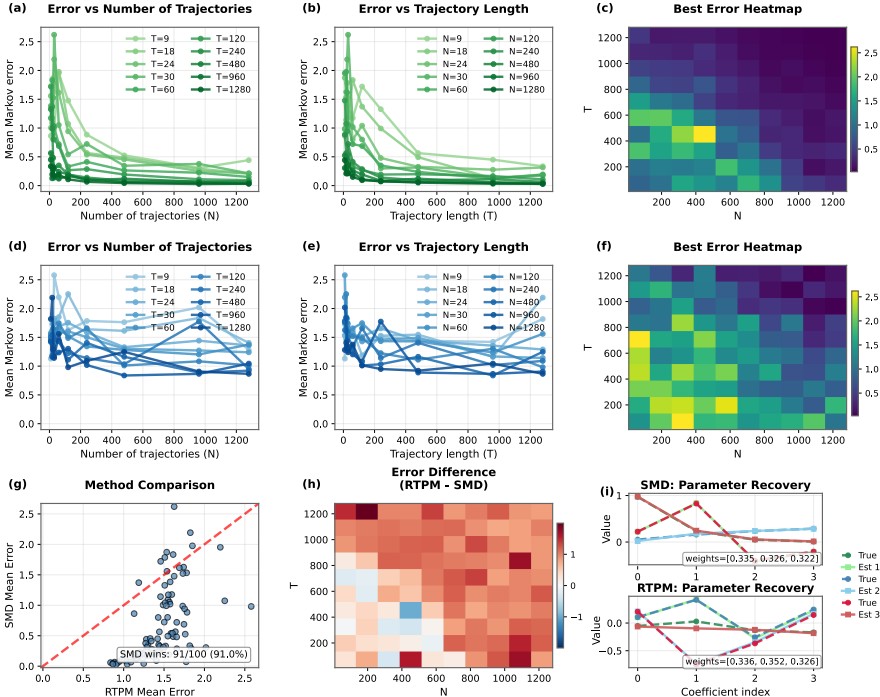

Figure 2: **Comparison of SMD-Tensor and RTPM-Tensor methods for MoLDS:** (a,b) Mean Markov parameter estimation errors of the SMD method decrease as the number of trajectories as $N$ and/or $T$ increase; (c) Heatmap of the best trial result across all $(N, T)$ configurations. (d-f) Corresponding results for the RTPM method. (g) Scatter plot comparing mean Markov errors of RTPM vs. SMD across configurations, with SMD outperforming in $91\%$ of cases. (h) Difference in mean Markov errors between RTPM and SMD (positive values indicate SMD performs better). (i) Example recovery for $N = T = 1280$, where SMD recovers both mixture weights and Markov parameters more accurately.

The first row of Figure 2 reports results of the SMD-Tensor method for a $K = 3$ mixture model with LDS dimensions $n = 2, m = p = 1$. The LDS parameters are randomly generated, with eigenvalues of $A$ constrained to lie inside the unit circle. We vary the trajectory length $T$ and the number of trajectories $N$ for each configuration. We run multiple independent trials, calculating the discrepancy between estimated and true Markov parameters. The second row of Figure 2 shows the result for the RTPM-Tensor method. It is noticed that the mean Markov parameter errors decrease with increasing trajectory length and number of trajectories for both methods, but the trend is more pronounced and

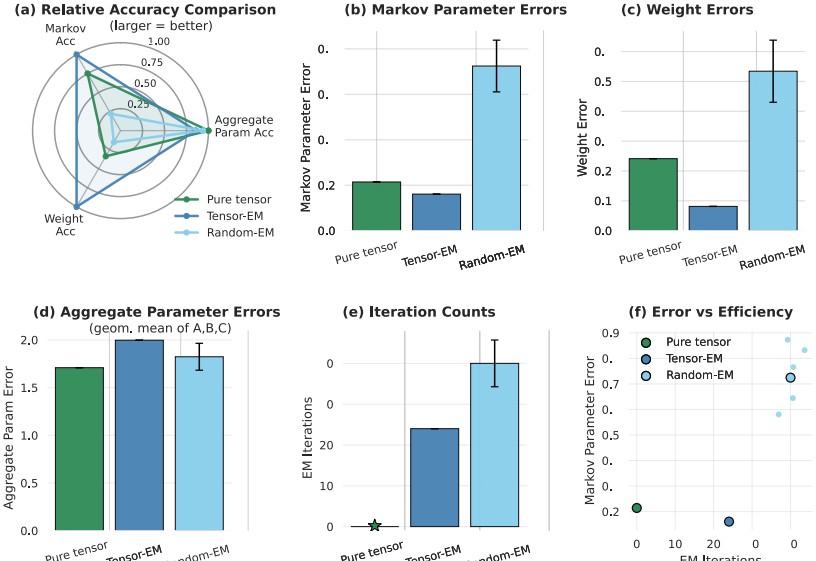

Figure 3: **Performance comparison of pure tensor, Tensor-EM, and random-initialized EM on simulated MoLDS:** (a) Relative accuracy radar plot across metrics of Markov parameter accuracy, weight accuracy, and aggregate parameter accuracy. (b-c) Tensor-EM achieves the lowest Markov parameter and weight errors, while random EM performs the worst and shows high variability. (d) reports aggregate parameter errors (geometric mean of $A, B, C$ errors). (e) Tensor-EM converges in far fewer iterations than random EM, highlighting its efficiency. (f) The error-efficiency plot shows that Tensor-EM combines low error with moderate iteration cost, yielding robust and accurate recovery.

stable for SMD, as reflected in the heat maps (Figure 2c,f). Across nearly all $(N, T)$ cases, SMD yields consistently lower errors (Figures 2g,h). In cases with larger $T$ and $N$, SMD achieves highly accurate recovery of both mixture weights and Markov parameters (Figure 2i), underscoring its robustness and reliability for the MoLDS setting.

## 4.2 TENSOR-EM IMPROVES ROBUSTNESS AND ACCURACY FOR COMPLEX SYNTHETIC MOLDS

In complex MoLDS settings with many components, purely tensor-based methods and randomly initialized EM often fail to achieve accurate recovery, either due to noisy parameter estimates or convergence to poor local optima. Our proposed Tensor-EM approach overcomes these limitations by combining globally consistent tensor initialization with EM-based refinement, resulting in more robust and accurate learning. We demonstrate these advantages in this section.

Figure 3 presents results on a simulated $K = 6$ MoLDS ($n = 3$, $m = p = 2$) with zero direct feedthrough ($D = 0$). Across metrics including Markov parameter errors, mixture weight errors, aggregate LDS parameter errors, and iteration counts, Tensor-EM consistently outperforms all baselines. It achieves substantially lower Markov and weight errors (panels b and c), while maintaining an aggregate parameter error comparable to the pure tensor solution (panel d).[2] At the same time, Tensor-EM converges in much fewer iterations than random EM (panel e), leading to a better error-efficiency tradeoff (panel f). The radar plot ( Figure 3a) summarizes these improvements, showing that Tensor-EM achieves the most balanced performance across all evaluation metrics. These results highlight that Tensor-EM effectively combines the strengths of tensor methods and EM and enables more reliable parameter recovery in synthetic MoLDS settings where standalone tensor and randomly initialized EM approaches often underperform.

---

[2]The aggregate parameter error, i.e., the geometric mean of $A, B, C$ errors, is a coarse summary: unlike Markov parameter errors, it may be less precise since $A, B, C$ can differ by similarity transformations without altering the underlying dynamics.

# 5 TENSOR-EM MOLDS PROVIDES RELIABLE AND INTERPRETABLE RECOVERY ON REAL-WORLD APPLICATIONS

We next apply the proposed MoLDS method to two real-world neural datasets.

## 5.1 AREA2 DATASET

We first analyze a neural dataset from a single recording session of a macaque performing center-out reaches, with neural activity recorded from the somatosensory cortex (Figure 4a). During the experiment, the monkey used a manipulandum to direct a cursor from the center of a display to one of eight targets, and neural activity was recorded from Brodmann's area2 of the somatosensory cortex (see (Chowdhury et al., 2020) for experimental details, and (Miller, 2022) for dataset location). The neural recordings consist of 65 single-unit spike times, which are converted to spike rates. In addition to neural data, the position of the monkey's hand, cursor position, force applied to the manipulandum, and hand velocity were recorded during the experiment. There are 8 directions in the task, and each direction has multiple trials of trajectories. For the MoLDS fitting, we extract the movement-related segment of each trial, defined as the window from 100 ms before to 500 ms after movement onset. The 65-dimensional

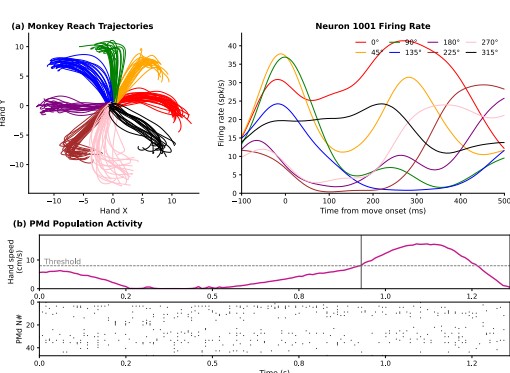

Figure 4: **Neural datasets overview:** (a) The Area2 Dataset contains neural trajectories from monkey primary somatosensory cortex; reach trajectories and firing rate from multiple neurons. (b) The PMd Dataset contains recordings from monkey dorsal premotor cortex; hand speed and neurons' rasters.

observations are reduced to 6-dimension by using the standard PCA, which explains more than 90 percent variance of neural activities. These PCs are then taken as outputs, while the hand velocity variables are taken as inputs. In addition, we also evaluate the dataset with 20 PCs using MoLDS, and consistent results are found (see Figure 11), which confirms the effectiveness of the Tensor-EM algorithm to fit a MoLDS model on this dataset.

We evaluate the MoLDS with Tensor-EM, Random-EM, and pure tensor methods using a standard train/validation/test split of the dataset (see full pipeline in App. E). For each hyperparameter setting (including $K$), models are trained on the training set and scored on the validation set using negative log-likelihood (NLL), one-step-ahead RMSE, and BIC. The model used for test-time analysis is the one minimizing validation BIC (we also report NLL/RMSE). For trial $i$ and component $k$, we compute a responsibility $r_{ik} \propto \exp(\ell_k^{(i)})$, where $\ell_k^{(i)}$ is the one-step Kalman log-likelihood under component $k$. For each movement direction, the dominant component is the $\arg\max_k$ of the mean responsibility across its trials.

Figure 5 summarizes the results. As shown in Figure 5a, the validation criteria consistently favor a 3-component MoLDS trained with Tensor-EM. Moreover, the one-step predictions $\hat{y}_t$ closely track observations $y$ on the test data as in Figure 5b. In Figure 5c, the dominant-component maps reveal three cross-trial clusters aligned with directions, and the usage fractions in Figure 5d quantify the prevalence of MoLDS components on the test set.

In this setup, we also compare the Tensor-EM MoLDS method with the supervised learning results of LDS, where we train (1) one LDS on all trials where the monkey reaches in a specific direction (per-dir LDS), and separately (2) a separate LDS fit on each trial regardless of reaching direction (single-trial LDS). Finally, we cluster the parameters of the different LDS's (per-dir or single-trial). See Figure 8 in the Appendix for a representation of the clustered parameters and more details. The per-direction LDS baseline closely matches the result of MoLDS with the Tensor-EM method rather than Random-EM in trial groupings (see Figure 5c), and the impulse responses are highly similar (see Figure 10 in the Appendix). We also train an SLDS in similar unsupervised way as MoLDS (no direction labels) and find that it does not yield meaningful cross-trial clusters here, while it is effective

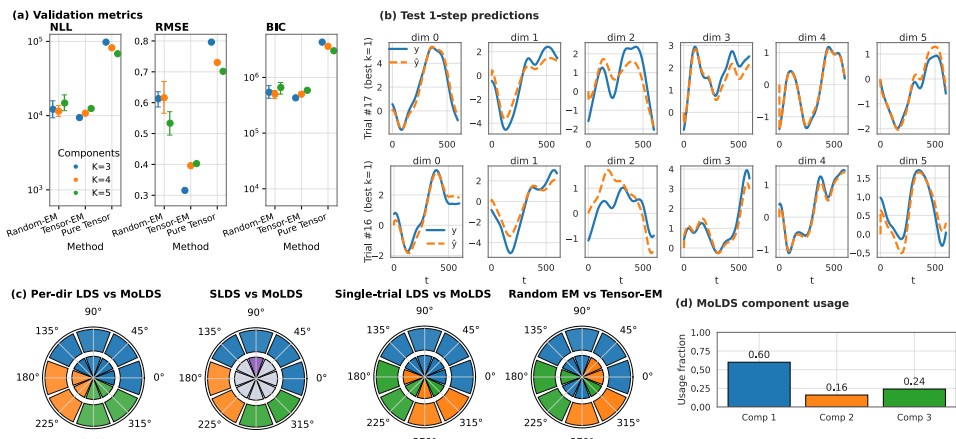

Figure 5: **MoLDS application on Area2 Dataset:** (a) Validation metrics of MoLDS for different $K$. (b) One-step predictions for two example trials using corresponding LDS components from the MoLDS selected by the lowest validation metrics. (c) Agreement between Tensor-EM MoLDS trial assignments (outer ring) and per-direction LDS, SLDS, single-trial LDS, Random-EM MoLDS clusters (inner rings in each polar plot); the SLDS-based method cannot provide meaningful cross-trial clusters. (d) Tensor-EM MoLDS component usage fractions on held-out test trials.

for possible within-trial regime switches (see Figure 9 in Appendix). These highlight MoLDS's strength in capturing between-trial heterogeneity. In addition, when compared with the Random-EM method, Tensor-EM offers a key advantage: the tensor initialization step yields a stable starting point that reduces variability across runs, leading to more consistent parameter recovery and recovered model structure as seen in Figures 5a-c.

## 5.2 PMD DATASET

We next apply our method to recordings from monkey dorsal premotor cortex (PMd) during sequential reaches (Figure 4a), in which trial-wise movement directions are *continuously distributed* over the circle. Full experimental details are in (Lawlor et al., 2018) and the dataset is provided in (Perich et al., 2018). The trials' reach direction spans the full polar range rather than clearly separated directions as in the Area2 dataset. For the PMd dataset, we similarly extract movement-related activity by taking a fixed window ($-100$ to $+500$ ms) around the movement onset for each trial. We analyze PMd activity with the 5-dimensional kinematic as inputs, i.e., (x-velocity, y-velocity, x-acceleration, y-acceleration, speed), and the first 16 PCs as outputs. Following the Area2 protocol, we train MoLDS across hyperparameters on the PMd training split and select the model by validation criteria (NLL/RMSE/BIC). Figure 6a reports validation results where the 4-component MoLDS trained with the Tensor-EM method has better performance. Figure 6b shows test one-step-ahead reconstructions of the corresponding component from this MoLDS fit.

To accommodate continuously distributed reach angles, we uniformly binned trials into 12 angular bins based on their reach angles. For each bin, we assigned a dominant component based on the highest mean responsibility. Figure 6c shows these bins colored by their dominant LDS component and overlays responsibility-weighted preferred-direction (PD) arrows for each component, where we also compare Tensor-EM and Random-EM MoLDS results. Figure 6d plots impulse-response magnitude across lags of each component, revealing component-specific gains and temporal decay; Appendix E.2 (Figure 14) further shows the decomposed responses across input channels. Figure 6e reports Tensor-EM MoLDS component usage fractions on held-out test trials. Overall, the mixture components of the PMd dataset specialize in different movement directions and exhibit distinct dynamical response profiles.

## 6 CONCLUSION

MoLDS provides an interpretable framework for modeling repeated, trajectory-level dynamics in heterogeneous data. In this work, we proposed a hybrid Tensor-EM approach for efficient and

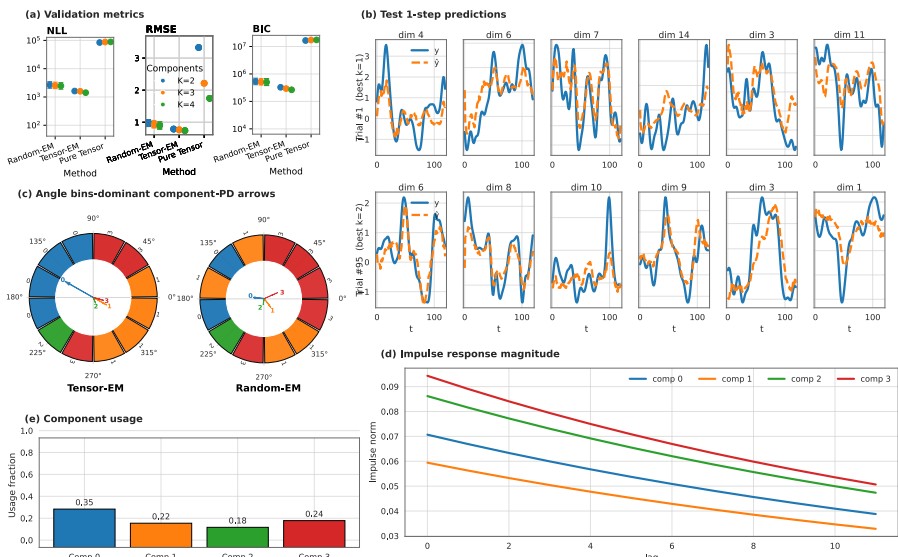

Figure 6: **MoLDS application on PMd Dataset:** (a) Validation metrics of MoLDS for different $K$ (NLL, RMSE, BIC). (b) One-step prediction on an example neural trial using the corresponding component from the validation-chosen MoLDS. (c) Angle bins: dominant component; PD arrows (responsibility-weighted). (d) Impulse response magnitude of Tensor-EM MoLDS components. (e) Tensor-EM MoLDS component usage fractions on held-out test trials.

reliable MoLDS inference. Specifically, we used SMD for tensor-based initialization, followed by Kalman filter–smoother EM refinement. This combination enables accurate parameter recovery and interpretable trajectory clustering in both synthetic benchmarks and real neural datasets. This hybrid framework makes MoLDS practically usable for large-scale, noisy datasets.

Our empirical evaluation focused on scenarios with linear dynamics: synthetic data generated from linear systems and neural data consisting of smoothed firing rates over selected movement windows. While these experiments validate the effectiveness of the proposed inference procedure, we have not explored cases with a high likelihood of model mismatch or strong nonlinearities. We envision a few extensions that can help broaden the applicability of MoLDS. First, nonlinear dynamics may be approximated by segmenting trajectories into shorter sessions that are locally well-approximated by linear dynamical systems. Second, linear components could be augmented with nonlinear features to capture moderate deviations from linear structure. In addition, future directions can include extending the framework to autonomous settings and developing theoretical guarantees under simplified assumptions.

More broadly, MoLDS is well-suited to various neuroscience tasks in which multiple experimental conditions share latent dynamical structure but differ in observable trajectories. For example, in sensory decision-making with different stimulus strengths, distinct conditions may reuse a small set of evidence-integration dynamics (Hanks et al., 2015). In context-switching tasks, neural populations in the prefrontal cortex may transition among a number of rule-dependent dynamical modes (Mante et al., 2013). In motor learning experiments, early, intermediate, and late learning stages may correspond to transitions among a small set of dynamical regimes (Shenoy et al., 2013). By developing a principled Tensor-EM learning and inference framework, this work establishes a practical and extensible foundation for applying MoLDS to complex neural datasets. Beyond neuroscience, the framework may also support broader applications, such as identifying clinically relevant behavioral dynamics (Bulteel et al., 2016) or uncovering cell-type-specific dynamical structure (Luecke et al., 2021).

## 7 ETHICS STATEMENT

Here, we aim to make methodological and neuroscientific insights, and do not note any negative societal or ethical implications.

## 8 Reproducibility Statement

Our work can be reproduced in a straightforward way. The datasets are provided in (Miller, 2022) and (Perich et al., 2018), with all pre-processing techniques detailed in (Chowdhury et al., 2020), (Lawlor et al., 2018), and in the Appendix of this work. Moreover, detailed algorithms and technical details are provided for each step of the learning and inference, with comprehensive pseudo-code for the implementation in the main text and the Appendix.

## 9 Acknowledgments

L. Gong was supported by the Swartz Foundation Postdoctoral Fellowship. S. Saxena was supported by the National Science Foundation grant No. 2350329 and the National Institutes of Health grant No. R34DA059718. The authors especially thank Maryann Rui and Dr. Munther Dahleh for helpful discussions.

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

## A  LLM USAGE STATEMENT

We used LLMs to assist with language editing in parts of the manuscript. All LLM's suggestions were manually verified and edited by the authors.

## B  TENSOR INITIALIZATION DETAILS

In this section, we provide a brief introduction to MoLDS's connection to MLR and the tensor-based moment method for MoLDS (see (Bakshi et al., 2023; Rui & Dahleh, 2025) for more details).

### B.1  MoLDS TO MLR REFORMULATION

The key insight enabling tensor methods for MoLDS is that any linear dynamical system can be equivalently represented through its *impulse response* (also called Markov parameters), which describes how the system responds to unit impulses in its inputs. This representation allows us to reformulate MoLDS as MLR, making algebraic moment-based methods applicable. We now briefly introduce this process following the formulation in (Rui & Dahleh, 2025).

**Step 1: Impulse response representation.**  For notational brevity, we assume an LDS with $m$-dimensional inputs and a scalar output ($p = 1$), and zero feedthrough ($D_k = 0$). The extension to multiple outputs ($p > 1$) is straightforward. The system parameters are $(A_k, B_k, C_k)$. Its impulse response, also called the sequence of Markov parameters, is defined by

$$g_k(j) \;=\; C_k A_k^{j-1} B_k \;\in\; \mathbb{R}^m, \qquad j = 1, 2, 3, \ldots,$$

so that $g_k(1) = C_k B_k$, $g_k(2) = C_k A_k B_k$, etc.

The sequence $\{g_k(j)\}$ captures the system's memory: $g_k(j)$ specifies how an input applied $j$ steps in the past influences the current output. Given an input sequence $\{u_t\}$, the output can be expressed as the infinite convolution

$$y_t = \sum_{j=1}^{\infty} g_k(j)\, u_{t-j} + \text{noise terms.}$$

In practice, this sum is truncated after $L$ terms, since $g_k(j)$ decays exponentially for stable systems. We denote the stacked first $L$ impulse responses by

$$g_k^{(L)} = [g_k(1)^\top, g_k(2)^\top, \cdots, g_k(L)^\top]^\top \in \mathbb{R}^{Lm}.$$

**Step 2: Lagged input construction.** To exploit the impulse response representation, we construct *lagged input vectors* that collect recent input history. For trajectory $i$ at time $t \geq L$, define

$$\bar{u}_{i,t} = [u_{i,t}^\top, u_{i,t-1}^\top, \ldots, u_{i,t-L+1}^\top]^\top \in \mathbb{R}^{Lm}.$$

This vector stacks the most recent $L$ inputs (the current input and the previous $L-1$), so that $\bar{u}_{i,t}$ has dimension $Lm$ and aligns with the stacked impulse responses $g_k^{(L)}$. With this construction, the truncated output becomes

$$y_{i,t} \approx \langle g_{k_i}^{(L)}, \bar{u}_{i,t} \rangle + \text{noise terms},$$

where $k_i$ denotes the (unknown) component generating trajectory $i$.

With the above input construction, consecutive lagged vectors $\bar{u}_{i,t}$ and $\bar{u}_{i,t+1}$ share $L-1$ input entries, which induces strong statistical dependence and complicates the estimation of higher-order moments. To mitigate this, we sub-sample the time indices with a stride $L$, i.e.,

$$t \in \{L, 2L, 3L, \ldots, T\}.$$

This construction ensures that the resulting lagged vectors are non-overlapping. Under standard input assumptions (e.g., i.i.d. or persistently exciting inputs), the sub-sampled vectors are approximately independent, which is crucial for consistent moment estimation.

**Step 4: Normalization and MLR formulation.** To apply tensor methods, the covariates must have unit variance. We therefore normalize each lagged input by the input standard deviation,

$$v_j = \frac{\bar{u}_{i,t}}{\sigma_u}, \qquad \sigma_u^2 = \mathbb{E}\big[\|u_t\|^2\big],$$

and flatten the dataset via the re-indexing map $(i, t) \mapsto j$. This yields the mixture of linear regressions (MLR) form

$$\tilde{y}_j = \langle v_j, \beta_{k_j} \rangle + \eta_j + \xi_j,$$

where

- $\tilde{y}_j = y_{i,t}$ is the observed output,
- $v_j \in \mathbb{R}^{Lm}$ is the normalized lagged input (covariate),
- $\beta_{k_j} = \sigma_u g_{k_j}^{(L)} \in \mathbb{R}^{Lm}$ is the scaled Markov parameter vector (regression coefficient),
- $k_j \in \{1, \ldots, K\}$ is the (unknown) component index,
- $\eta_j$ captures process and observation noise,
- $\xi_j$ accounts for truncation error from ignoring impulse responses beyond lag $L$.

**Step 5: Mixture structure.** Since each trajectory originates from one of $K$ latent LDS components with probabilities $\{p_k\}_{k=1}^K$, the regression model inherits the same mixture structure:

$$\mathbb{P}[k_j = k] = p_k.$$

Thus the learning task reduces to recovering the mixture weights $\{p_k\}$ and regression vectors $\{\beta_k\}$ from the dataset $\{(v_j, \tilde{y}_j)\}$. Once these regression parameters are estimated, the corresponding state-space models $(A_k, B_k, C_k)$ can be reconstructed via the Ho-Kalman realization algorithm (Oymak & Ozay, 2019).

This reformulation is crucial: it transforms the original problem of identifying a mixture of LDSs into the algebraic problem of learning an MLR model, for which polynomial-time tensor methods with identifiability and sample-complexity guarantees are available (Rui & Dahleh, 2025).

### B.2 MOMENT CONSTRUCTION AND WHITENING

Let $d = Lm$ denote the dimension of the lagged input vectors. We partition the sample indices into two disjoint subsets $\mathcal{N}_2$ and $\mathcal{N}_3$ for constructing second- and third-order moments, respectively as in (Rui & Dahleh, 2025). The empirical moments are defined as

$$M_2 = \frac{1}{2|\mathcal{N}_2|} \sum_{j \in \mathcal{N}_2} \tilde{y}_j^2 (v_j \otimes v_j - I_d), \qquad M_3 = \frac{1}{6|\mathcal{N}_3|} \sum_{j \in \mathcal{N}_3} \tilde{y}_j^3 (v_j^{\otimes 3} - \mathcal{E}(v_j)),$$

where $I_d$ is the $d \times d$ identity and

$$\mathcal{E}(v) = \sum_{r=1}^{d} \big( v \otimes e_r \otimes e_r + e_r \otimes v \otimes e_r + e_r \otimes e_r \otimes v \big),$$

with $e_r$ the $r$-th standard basis vector in $\mathbb{R}^d$. These corrections ensure that the resulting tensors are centered and symmetric.

At the population level, these moments satisfy

$$\mathbb{E}[M_2] = \sum_{k=1}^{K} p_k \, \beta_k \beta_k^\top, \qquad \mathbb{E}[M_3] = \sum_{k=1}^{K} p_k \, \beta_k^{\otimes 3},$$

so they encode the regression vectors $\{\beta_k\}$ and mixture weights $\{p_k\}$.

To obtain an orthogonally decomposable form, we perform whitening. Let $M_2 \approx U\Sigma U^\top$ be the rank-$K$ eigendecomposition, and define

$$W = U_{(:,1:K)} \, \Sigma_{1:K}^{-1/2},$$

so that $W^\top M_2 W \approx I_K$. Applying $W$ along each mode of $M_3$ yields the whitened tensor

$$\widehat{T} = M_3(W, W, W) = \sum_{k=1}^{K} p_k \, \alpha_k^{\otimes 3}, \qquad \alpha_k := W^\top \beta_k.$$

This tensor is symmetric and orthogonally decomposable. We then apply Simultaneous Matrix Diagonalization (SMD) (B.3) (Kuleshov et al., 2015) to recover $\{\alpha_k, p_k\}$. Finally, we unwhiten to obtain $\beta_k = W^{-\top}\alpha_k$, and recover the state-space parameters $(A_k, B_k, C_k)$ via Ho-Kalman realization (Oymak & Ozay, 2019).

## B.3 SIMULTANEOUS MATRIX DIAGONALIZATION

This section provides the Simultaneous Matrix Diagonalization (SMD) method for tensor decomposition (Kuleshov et al., 2015) in Algorithm 2. SMD recovers tensor components by reducing the problem to joint matrix diagonalization, exploiting linear algebraic structure to recover all components simultaneously rather than sequentially as in RTPM.

**Jacobi Joint Diagonalization (JJD).** Given matrices $\{M_\ell\}_{\ell=1}^{L_0}$, JJD finds an orthogonal matrix $U$ such that $U^\top M_\ell U$ is as close to diagonal as possible for all $\ell$ simultaneously. We use the Jacobi rotation-based algorithm that iteratively applies Givens rotations to minimize the off-diagonal Frobenius norm $\sum_\ell \|U^\top M_\ell U - \mathrm{diag}(U^\top M_\ell U)\|_F^2$. Convergence is declared when the relative change in this objective falls below $10^{-8}$.

---

**Algorithm 2** Simultaneous Matrix Diagonalization (SMD) for Tensor Decomposition

---

**Require:** Noisy symmetric tensor $\widehat{T} \in \mathbb{R}^{K \times K \times K}$; number of random probes $L_0 \geq 2$.
**Ensure:** Factor estimates $\{\hat{\alpha}_i\}_{i=1}^{K}$ and weights $\{\hat{p}_i\}_{i=1}^{K}$ such that $\widehat{T} \approx \sum_{i=1}^{K} \hat{p}_i \, \hat{\alpha}_i^{\otimes 3}$.

    **Stage 1: random projections $\to$ simultaneous diagonalization**
1: Sample $\{w_\ell\}_{\ell=1}^{L_0}$ i.i.d. from the unit sphere $\mathbb{S}^{K-1}$.
2: Form projected matrices $\mathcal{M}^{(0)} \leftarrow \{\widehat{T}(I, I, w_\ell)\}_{\ell=1}^{L_0}$, where $\widehat{T}(I, I, w) = \sum_{r=1}^{K} w_r \, \widehat{T}(:, :, r)$.
3: Compute an approximate joint diagonalizer via **JJD**: $U^{(0)} \leftarrow \mathrm{JJD}(\mathcal{M}^{(0)})$.
4: Set $V^{(0)} \leftarrow (U^{(0)})^{-1}$.
    **Stage 2: inverse-guided projections $\to$ refinement**
5: Build $\mathcal{M}^{(1)} \leftarrow \{\widehat{T}(I, I, v_i^{(0)})\}_{i=1}^{K}$, where $v_i^{(0)}$ is the $i$-th column of $V^{(0)}$.
6: Refine with **JJD**: $U^{(1)} \leftarrow \mathrm{JJD}(\mathcal{M}^{(1)})$.
7: Let $\hat{\alpha}_i$ be the $i$-th column of $U^{(1)}$ and normalize to $\|\hat{\alpha}_i\|_2 = 1$.
8: **for** $i = 1$ to $K$ **do**
9:     $\hat{p}_i \leftarrow \langle \widehat{T}, \, \hat{\alpha}_i \otimes \hat{\alpha}_i \otimes \hat{\alpha}_i \rangle$.
10: **end for**
11: **return** $\{\hat{\alpha}_i\}_{i=1}^{K}, \{\hat{p}_i\}_{i=1}^{K}$.

---

### B.4 TENSOR INITIALIZATION ALGORITHM

With the MLR reformulation and the SMD tensor method, the full algorithm of tensor initialization for MoLDS can be provided now.

---

**Algorithm 3** Tensor Initialization for MoLDS

---

**Require:** Trajectories $\{(u_{i,0:T_i-1}, y_{i,0:T_i-1})\}_{i=1}^N$, truncation $L$, LDS order $n$, #components $K$
**Ensure:** Estimates of mixture weights $\hat{p}_{1:K}$ and LDS params $\{(\hat{A}_k, \hat{B}_k, \hat{C}_k, \hat{D}_k)\}_{k=1}^K$
    **(A) MoLDS → MLR design & sub-sampling**
 1: **for** $i = 1\!:\!N$ **do**
 2:     **for** $t \in \{L, 2L, \ldots, T_i\}$ **do**
 3:         $\bar{u}_{i,t} \leftarrow [u_{i,t}^\top, u_{i,t-1}^\top, \ldots, u_{i,t-L+1}^\top]^\top \in \mathbb{R}^{Lm}$
 4:         Append sample $(v_j, \tilde{y}_j)$ with $v_j \leftarrow \bar{u}_{i,t}/\sigma_u$, $\tilde{y}_j \leftarrow y_{i,t}$
 5:     **end for**
 6: **end for**
 7: Partition samples: $\{1, \ldots, M\} = \mathcal{N}_2 \,\dot\cup\, \mathcal{N}_3$ where $M = \sum_i \lfloor T_i/L \rfloor$
    **(B) Moment construction**
 8: $M_2 \leftarrow \frac{1}{2|\mathcal{N}_2|}\sum_{j \in \mathcal{N}_2} \tilde{y}_j^2 \,(v_j \otimes v_j - I_d)$ where $d = Lm$
 9: $M_3 \leftarrow \frac{1}{6|\mathcal{N}_3|}\sum_{j \in \mathcal{N}_3} \tilde{y}_j^3 \,(v_j^{\otimes 3} - \mathcal{E}(v_j))$
    **(C) Symmetric whitening and tensor formation**
10: Compute rank-$K$ SVD: $M_2 \approx U\Sigma U^\top$, set $W \leftarrow U_{(:,1:K)}\Sigma_{1:K}^{-1/2}$
11: Form whitened tensor: $\widehat{T} \leftarrow M_3(W, W, W) \in \mathbb{R}^{K \times K \times K}$
    **(D) Tensor decomposition & recovery**
12: Apply SMD to $\widehat{T}$ to recover $\{\hat{\alpha}_k, \hat{p}_k\}_{k=1}^K$ (Appendix Alg 2)
13: Unwhiten: $\hat{\beta}_k \leftarrow W^{-\top}\hat{\alpha}_k$ and recover Markov params $\widehat{g}_k^{(L)} \leftarrow \hat{\beta}_k/\sigma_u$
    **(E) State-space realization**
14: **for** $k = 1\!:\!K$ **do**
15:     Build Hankel matrix from $\{\widehat{g}_k^{(h)}\}_{h=1}^L$ and apply Ho-Kalman algorithm
16:     Recover state-space parameters $(\hat{A}_k, \hat{B}_k, \hat{C}_k, \hat{D}_k)$
17: **end for**
18: **return** $\{\hat{p}_k\}_{k=1}^K$ and $\{(\hat{A}_k, \hat{B}_k, \hat{C}_k, \hat{D}_k)\}_{k=1}^K$

---

Under standard identifiability and excitation conditions, RTPM-based tensor decomposition provably recovers the parameters $\{\beta_k\}$ with finite-sample guarantees (Rui & Dahleh, 2025). Simultaneous matrix diagonalization (SMD) enjoys analogous guarantees in the orthogonal-tensor setting and, in practice, tends to be more numerically stable and noise-robust (Kuleshov et al., 2015). These advantages lead to improved parameter estimates during the tensor initialization stage (Figure 2, main paper), which in turn can provide a stronger starting point for the subsequent EM refinement.

## C  EM FOR MoLDS: COMPLETE TECHNICAL DETAILS

### C.1  OVERVIEW AND STRUCTURE

This appendix provides complete technical details for our EM formulation for MoLDS. The EM algorithm for MoLDS alternates between two phases: (i) an E-step that computes trajectory-wise responsibilities via Kalman filter likelihoods and extracts sufficient statistics via Kalman smoothing, and (ii) an M-step that updates all parameters using closed-form maximum likelihood estimates from responsibility-weighted statistics. The algorithm is detailed in Algorithm 4.

### C.2  COMPLETE EM ALGORITHM

The EM procedure operates on trajectories $\{(u_{i,0:T_i-1}, y_{i,0:T_i-1})\}_{i=1}^N$ with current parameter estimates $\hat{\theta}^{(t)} = \{(\hat{p}_k, \hat{A}_k, \hat{B}_k, \hat{C}_k, \hat{D}_k, \hat{Q}_k, \hat{R}_k)\}_{k=1}^K$ at iteration $t$.

**E-step Computations.** For each trajectory $i$ and component $k$, we compute:

$$\ell_{i,k} = \log p\left(y_{i,0:T_i-1} \mid u_{i,0:T_i-1}, \hat{\theta}_k^{(t)}\right), \tag{8}$$

$$\alpha_{i,k} = \log \hat{p}_k^{(t)} + \ell_{i,k}, \tag{9}$$

$$\gamma_{i,k} = \frac{\exp(\alpha_{i,k})}{\sum_{r=1}^K \exp(\alpha_{i,r})}, \quad \sum_{k=1}^K \gamma_{i,k} = 1. \tag{10}$$

The likelihood $\ell_{i,k}$ is computed via the Kalman filter, while responsibilities $\gamma_{i,k}$ use the log-sum-exp trick for numerical stability. Subsequently, the Kalman smoother computes per-trajectory sufficient statistics $S_{i,k}$, which are aggregated as:

$$S_k = \sum_{i=1}^N \gamma_{i,k} S_{i,k}. \tag{11}$$

**M-step Updates.** Mixture weights and LDS parameters are updated via:

$$\hat{p}_k^{(t+1)} = \frac{1}{N} \sum_{i=1}^N \gamma_{i,k}, \tag{12}$$

$$\hat{\theta}_k^{(t+1)} = \text{MLE-LDS}(S_k), \tag{13}$$

where the closed-form LDS parameter updates are derived in C.3.

**Convergence.** The algorithm monitors the observed-data log-likelihood:

$$\mathcal{L}^{(t+1)} = \sum_{i=1}^N \log \sum_{k=1}^K \hat{p}_k^{(t+1)} p\left(y_{i,0:T_i-1} \mid u_{i,0:T_i-1}, \hat{\theta}_k^{(t+1)}\right), \tag{14}$$

and terminates when the relative improvement falls below the threshold $\varepsilon$.

---

**Algorithm 4** EM Refinement for MoLDS

---

**Require:** Trajectories $\{(u_{i,0:T_i-1}, y_{i,0:T_i-1})\}_{i=1}^N$; initial parameters $\hat{\theta}^{(0)}$; max iterations EM_max; tolerance $\varepsilon$
**Ensure:** Refined mixture weights and LDS parameters
 1: **for** iter $= 1$ to EM_max **do**
    **E-step: Compute responsibilities and sufficient statistics**
 2:    **for** $i = 1 : N$ **do**
 3:        **for** $k = 1 : K$ **do**
 4:            Compute $\ell_{i,k}$ via Kalman filter on $(u_{i,0:T_i-1}, y_{i,0:T_i-1})$ using $\hat{\theta}_k^{(\text{iter}-1)}$
 5:            Set $\alpha_{i,k} \leftarrow \log \hat{p}_k^{(\text{iter}-1)} + \ell_{i,k}$
 6:            Run Kalman smoother to compute sufficient statistics $S_{i,k}$
 7:        **end for**
 8:        Compute $\text{lse}_i \leftarrow \log\sum_{r=1}^K \exp(\alpha_{i,r})$ and responsibilities $\gamma_{i,k} \leftarrow \exp(\alpha_{i,k} - \text{lse}_i)$
 9:    **end for**
10:    Aggregate responsibility-weighted statistics: $S_k \leftarrow \sum_{i=1}^N \gamma_{i,k} S_{i,k}$ for each $k$
    **M-step: Update all parameters**
11:    Update mixture weights: $\hat{p}_k^{(\text{iter})} \leftarrow \frac{1}{N} \sum_{i=1}^N \gamma_{i,k}$ for all $k$
12:    **for** $k = 1 : K$ **do**
13:        Update LDS parameters $(\hat{A}_k, \hat{B}_k, \hat{C}_k, \hat{D}_k, \hat{Q}_k, \hat{R}_k)$ from $S_k$ (see C.3)
14:    **end for**
    **Convergence check**
15:    Compute observed log-likelihood: $\mathcal{L}^{(\text{iter})} \leftarrow \sum_{i=1}^N \text{lse}_i$
16:    **Stop if** $(\mathcal{L}^{(\text{iter})} - \mathcal{L}^{(\text{iter}-1)})/|\mathcal{L}^{(\text{iter}-1)}| < \varepsilon$
17: **end for**
18: **Return** $\{\hat{p}_k, (\hat{A}_k, \hat{B}_k, \hat{C}_k, \hat{D}_k, \hat{Q}_k, \hat{R}_k)\}_{k=1}^K$

---

## C.3   Closed-form M-step Updates

The M-step updates follow standard LDS maximum likelihood estimation (Ghahramani & Hinton, 1996). For simplicity, we present the case $D_k = 0$ (no direct feedthrough) and drop component and iteration indices $(k, t)$ for readability:

We solve the normal equations with optional ridge regularization $\lambda > 0$ to obtain system matrices for each component.

$$[A \quad B] = [S_{xx^-} \quad S_{xu^-}] \left( \begin{bmatrix} S_{x^-x^-} & S_{u^-x^-}^\top \\ S_{u^-x^-} & S_{u^-u^-} \end{bmatrix} + \lambda I \right)^{-1}, \tag{15}$$

$$C = S_{yx} \left( S_{xx} + \lambda I \right)^{-1}. \tag{16}$$

And the noise covariances are

$$R = \frac{1}{T} \left( S_{yy} - C S_{yx}^\top - S_{yx} C^\top + C S_{xx} C^\top \right), \tag{17}$$

$$Q = \frac{1}{N} \Big( S_{xx,\text{curr}} - A S_{xx^-}^\top - B S_{xu^-}^\top - (A S_{xx^-}^\top + B S_{xu^-}^\top)^\top \tag{18}$$

$$+ A S_{x^-x^-} A^\top + A S_{u^-x^-}^\top B^\top + B S_{u^-x^-} A^\top + B S_{u^-u^-} B^\top \Big). \tag{19}$$

Here, all $S_{\cdot,\cdot}$ are the sufficient statistics.

## C.4   Computational Complexity

Each EM iteration requires $O(NKn^3T)$ operations:

- Kalman filtering: $O(n^3T)$ per trajectory-component pair, total $O(NKn^3T)$
- Kalman smoothing: $O(n^3T)$ per trajectory-component pair, total $O(NKn^3T)$
- M-step updates: $O(Kn^3)$ for matrix inversions

Tensor initialization significantly reduces the iteration count compared to random initialization, which substantially improves computational efficiency. The tensor initialization phase requires $O(d^3)$ operations where $d = Lm$ is the MLR dimension, making it negligible in the whole Tensor-EM pipeline.

# D   Initializing $(Q, R)$ After Tensor Initialization

The tensor stage yields $\{\hat{A}_z, \hat{B}_z, \hat{C}_z, \hat{p}_z\}_{z=1}^K$ but not the noise covariances $\{\hat{Q}_z, \hat{R}_z\}$. We initialize $(Q_z, R_z)$ from data for each component $z$ as follows.

*Step 1 (labels/weights; optional).* Assign labels by selecting, for each trajectory $i$, the component $z$ with the smallest one-step prediction MSE under $(\hat{A}_z, \hat{B}_z, \hat{C}_z)$, and set $w_{i,z} = 1$ for that component and $w_{i,r} = 0$ for all $r \neq z$.

*Step 2 (state back-projection).* For each $z$ and trajectory $i$, decode provisional latents with a ridge pseudo-inverse:

$$\hat{x}_{i,t}^{(z)} = \left( \hat{C}_z^\top \hat{C}_z + \lambda I \right)^{-1} \hat{C}_z^\top (y_{i,t} - \hat{D}_z u_{i,t}),$$

where we recommend $\lambda = 10^{-6} \max(\text{diag}(\hat{C}_z^\top \hat{C}_z))$ for numerical stability.

*Step 3 (residual covariances).* Define

$$\eta_{i,t}^{(z)} = \hat{x}_{i,t+1}^{(z)} - \hat{A}_z \hat{x}_{i,t}^{(z)} - \hat{B}_z u_{i,t}, \qquad \varepsilon_{i,t}^{(z)} = y_{i,t} - \hat{C}_z \hat{x}_{i,t}^{(z)} - \hat{D}_z u_{i,t}.$$

With $T_i^{\text{tr}} = T_i - 1$, set

$$\hat{Q}_z^{(0)} = \frac{\sum_i w_{i,z} \sum_{t=0}^{T_i-2} \eta_{i,t}^{(z)} \eta_{i,t}^{(z)\top}}{\sum_i w_{i,z} T_i^{\text{tr}}}, \qquad \hat{R}_z^{(0)} = \frac{\sum_i w_{i,z} \sum_{t=0}^{T_i-1} \varepsilon_{i,t}^{(z)} \varepsilon_{i,t}^{(z)\top}}{\sum_i w_{i,z} T_i}.$$

*Step 4 (positive semidefinite projection).* Symmetrize and project to the positive semidefinite cone:

$$\hat{Q}_z^{(0)} \leftarrow \Pi_{\succeq 0}\big(\tfrac{1}{2}(\hat{Q}_z^{(0)} + \hat{Q}_z^{(0)\top})\big), \quad \hat{R}_z^{(0)} \leftarrow \Pi_{\succeq 0}\big(\tfrac{1}{2}(\hat{R}_z^{(0)} + \hat{R}_z^{(0)\top})\big),$$

where $\Pi_{\succeq 0}(M)$ projects matrix $M$ to the nearest positive semidefinite matrix via eigendecomposition: if $M = U\Lambda U^\top$, then $\Pi_{\succeq 0}(M) = U \max(\Lambda, 0)U^\top$.

This yields $(\hat{Q}_z^{(0)}, \hat{R}_z^{(0)})$ for each component $z$, providing a stable initialization for the first E-step; subsequent EM iterations refine $(Q_z, R_z)$ from responsibility-weighted sufficient statistics.

## E  TENSOR-EM MoLDS APPLICATION ON NEURAL DATASET DETAILS

### E.1  AREA2 DATASET

**Dataset introduction.**  We evaluate our methods on a neural population dataset recorded from the motor cortex (Area2) of a rhesus macaque. The task is a center-out reaching paradigm where the subject applies force to a manipulandum in response to cues. Neural activity was recorded with a Utah array, and spike counts were extracted from thresholded multiunit activity. The dataset provides simultaneously measured behavioral covariates (applied force, hand kinematics) alongside the neural recordings.

**Data preparation for MoLDS.**  Following the Neural Latents Benchmark preprocessing, we first apply principal component analysis (PCA) to reduce the neural activity to $p$ latent dimensions (here we use $p = 6, 20$). The behavioral force signals (6D) are used as exogenous inputs. For MoLDS training, we constructed input-output trials of the form $(u_t, y_t)$, where $u_t \in \mathbb{R}^6$ corresponds to force inputs and $y_t \in \mathbb{R}^p$ are PCA-reduced neural outputs. Each reaching direction corresponds to a separate trial. We split the dataset into non-overlapping `train`, `val`, and `test` sets, ensuring balanced coverage across each direction.

We checked the variance explained by PCA and found that the first six PCs capture over $90\%$ of the total variance for each direction, while the first 20 PCs explain nearly all of it. This indicates that PCA preserves the essential structure of the neural activity. The following figure illustrates this.

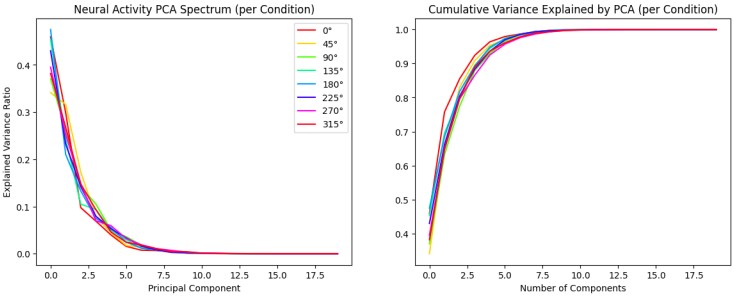

Figure 7: Area2 neural activity PCA spectrum.

**Tensor-EM MoLDS train/val/test setup.**  For model initialization, we computed input-output moment tensors from the training data and applied SMD to obtain globally consistent estimates of system parameters and mixture weights. These estimates were used to initialize a full Kalman filter-smoother EM refinement, which updates all LDS parameters in closed form. We trained MoLDS models with different numbers of mixture components $K \in 3, 4, 5$, latent dimension $n \in 3, 4, 5$, and lag parameters $L \in 16, 24$. Validation data is used for model selection by comparing negative log-likelihood (NLL), root mean squared error (RMSE), and Bayesian Information Criterion (BIC).

- **NLL:** sum of Gaussian-innovation log-likelihoods.
- **RMSE:** root mean squared error of one-step predictions $\hat{y}_t$ vs. $y_t$.
- **BIC:** $\mathrm{BIC} = -2\,\mathrm{NLL} + p_\theta \log N_{\mathrm{obs}}$, where $p_\theta$ is the total parameter count and $N_{\mathrm{obs}}$ the number of observed outputs.

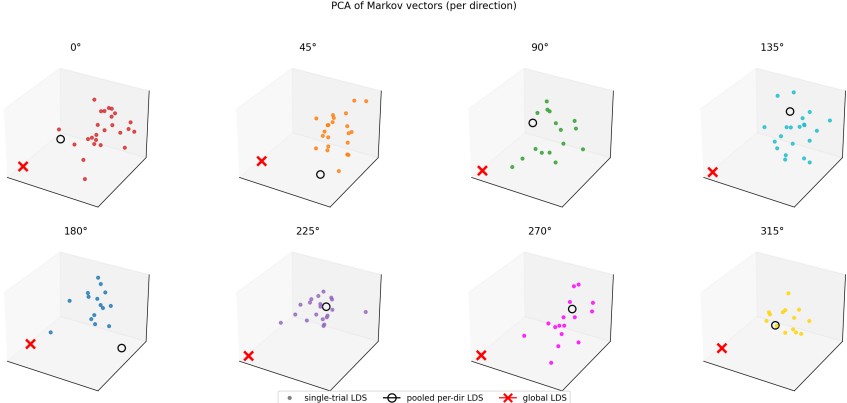

Figure 8: Markov parameters representation of LDS models from different fitting methods.

The best configuration was tested on the held-out `test` set.

**Per-trial responsibilities.** We compute responsibilities for each trial under component $k$

$$r_{ik} \propto \exp(\ell_k^{(i)}), \qquad \sum_{k=1}^{K} r_{ik} = 1,$$

where $\ell_k^{(i)}$ is the one-step Kalman log-likelihood of trial $i$ under component $k$. Global usage is summarized by $\text{usage}_k = \frac{1}{N} \sum_i r_{ik}$.

**Analyses.**

1. **Validation metrics.** We plot BIC/NLL/RMSE across $K$ and initializations (Tensor vs. Random) to assess stability and choose capacity.

2. **Representative predictions.** We visualize $\hat{y}_t$ vs. $y_t$ for representative test trials (all $p$ outputs).

3. **Direction organization & component usage.** In a polar wheel with one wedge per discrete direction, we color each wedge by the dominant component (highest mean $r_{ik}$ among trials in that wedge).

4. **Markov response comparison.** For each component, we calculate $g_k(\tau) = C_k A_k^\tau B_k \in \mathbb{R}^{p \times m}$, $\tau = 0, \ldots, L-1$ (omit $D_k$ for clarity). Plot $\|g_k(\tau)\|_F$ for all components on a single axis to compare gain/decay.

5. **Global geometry of dynamics.** Vectorize all $\{g_k(\tau)\}_{k,\tau}$, run PCA across that set, and scatter the first two PCs; use color to encode lag $\tau$ and marker/edge to encode component. Components typically form distinct, smoothly evolving trajectories.

**Supervised per-direction LDS fitting baseline.** Because Area2 uses *discrete* directions, we also fit a single LDS per direction using the same observation space. We compute each supervised model's impulse response, cluster the per-direction vectors, and align clusters to MoLDS components via a Hungarian assignment on centroid distances. A polar plot (inner = supervised clusters, outer = MoLDS predictions) confirms that unsupervised MoLDS recovers direction-consistent dynamics.

**Single-trial LDS fitting baseline.** In addition, we also fit an LDS for each single trial. We compute the average LDS impulse response in each direction and compare the obtained clusters as compared to the per-direction LDS. Following the same procedure as in per-dir LDS fitting, we compare the clusters of single-trial LDS fitting with MoLDS components as well. We have examined the trial-to-trial variability of the Area2 dataset by comparing their full Markov parameter vectors between single-trial, perd-dir, and single global LDS fits. As shown in Figure 8, by evaluating the first three PCs of the Markov vectors, single-trial LDSs within each direction form compact clusters, and the

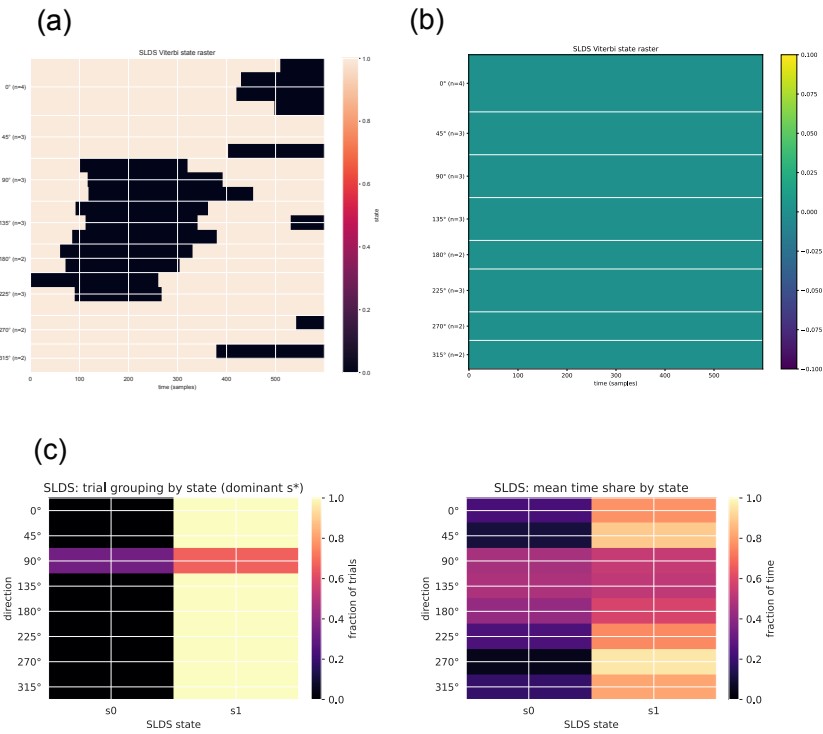

Figure 9: **SLDS result:** Inferred switch states for all test trials with mild (a) and strong stickiness (b). (a) Group trials and mean time by states with the mildly sticky SLDS fit.

pooled per-direction LDS lies near the cluster center for most directions. In contrast, these single-trial LDS are very far from a global LDS fitted on all trials. This indicates that within-direction variability is mild, and the per-direction LDS provides a stable summary of the underlying dynamics rather than an oversimplification. We also performed clustering directly on independently fitted single-trial LDSs. These LDSs naturally cluster by movement direction, even without pooling, demonstrating that direction-specific dynamical signatures are already present at the single-trial level. Moreover, by comparing with the per-dir LDS and MoLDS fit, these single-trial LDS fit-based clusters are very similar, with only two directions being switched between clusters (see Figure 12). Together, these analyses show that while the per-direction LDS is not a perfect "gold standard," it is a reasonable and stable reference, and our comparisons to MoLDS capture meaningful structure in the data.

**SLDS comparison.** We fit switching LDS (SLDS) models with $S \in 1, 2, 3$ discrete states to the training set (Gaussian emissions, 6-D force inputs, mild or strong sticky transitions). Validation NLL/BIC selected $S = 2$ SLDS. We then decoded the test set with Viterbi and summarized the inferred state sequences (Figure 9). Panel (a) shows that within-trial switches are infrequent and tend to occur in similar temporal regions across trials. In panel (b, left), grouping test trials by their dominant state $s^*$ reveals that the same state dominates most directions, with only a mild departure around dir-90. The mean time-share view (b, right) tells the same story: state composition looks broadly similar across directions. Consistent with this, a "collapsed" SLDS that uses only each trial's dominant state achieves nearly the same predictive metrics as the fully segmented model, indicating that within-trial switching adds little in this dataset. These results support our interpretation that the primary structure is between trials (by direction), not within trials.

**More results.**

**Impulse response comparison.** We compare the direction-specific impulse response (Markov parameter) curves of the assigned MoLDS component and the corresponding single LDS fit for that direction, which shows high similarity (see Figure 10 where titles report cosine similarity).

**Results of MoLDS fitting using 20 PCs.** We also applied the MoLDS method on the Area2 dataset with 20 PCs, and found consistent results for the 6-PC fitting model as shown in Figure 11.

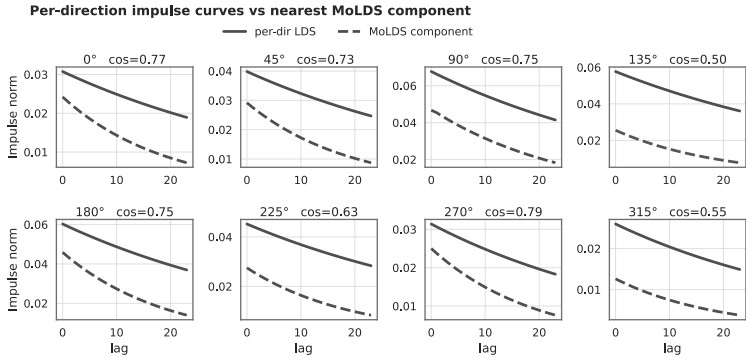

Figure 10: Impulse response comparison between MoLDS components and per-dir LDS fit.

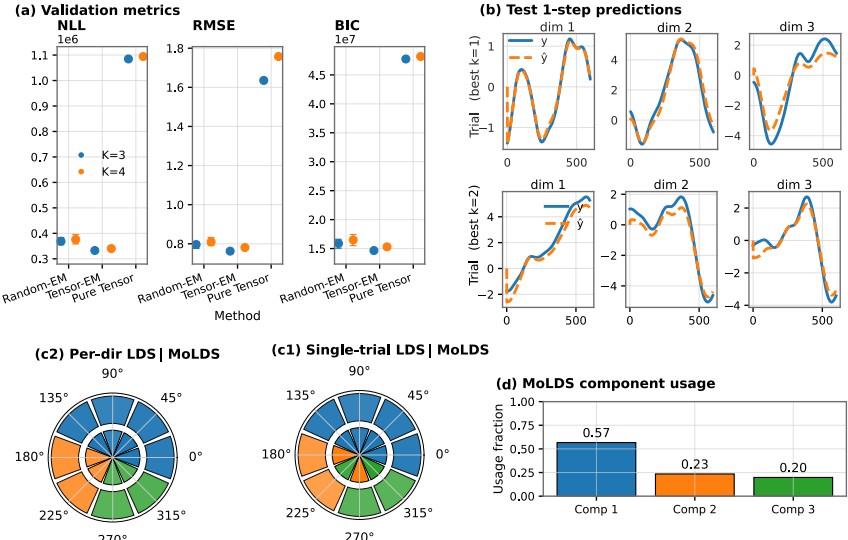

Figure 11: **MoLDS application on Area2 with 20 PCs:** results are consistent with those presented in the main text.

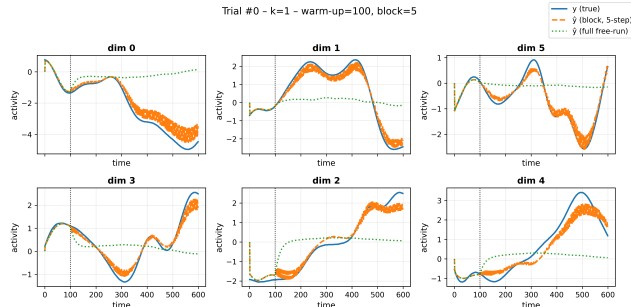

Figure 12: Multi-step forward predictions of MoLDS on one trial of Area2 data: 5-step vs fully free run after 100ms warm-up.

Figure 13: Area2 MoLDS fit dominant and second component analysis.

**Multi-step prediction** In addition to the one-step-ahead predictions presented in the main paper, we also inspected multi-step and free-run predictions. A representative example is provided in Figure 12. The model follows the true neural trajectory for several steps before diverging as it runs forward without correction, which is the characteristic and expected pattern for linear dynamical systems. This behavior is consistent with LDS-based modeling and does not affect our use of MoLDS for identifying shared local dynamical structure across trials.

**Responsibility distribution.** We analyzed the full responsibility distribution of MoLDS components on held-out test trials to assess whether non-dominant components carry substantial probability. Across all trials, the dominant responsibility is very large $0.976$, while the second-largest responsibility is very small $0.023$. This pattern is consistent across directions as shown in Figure 13.

### E.2  PMD DATASET

**Dataset introduction.**  The dorsal premotor cortex (PMd) dataset contains single-trial reaches, where time-aligned kinematics (position/velocity/acceleration) and multi-unit spikes, plus a reach direction angle $\theta \in (-\pi, \pi]$ are stored. For MoLDS, the input and observation are taken as

$$U_t = \begin{bmatrix} v_x, \, v_y, \, a_x, \, a_y, \, \text{speed} \end{bmatrix} \in \mathbb{R}^5, \qquad Y_t \in \mathbb{R}^{16},$$

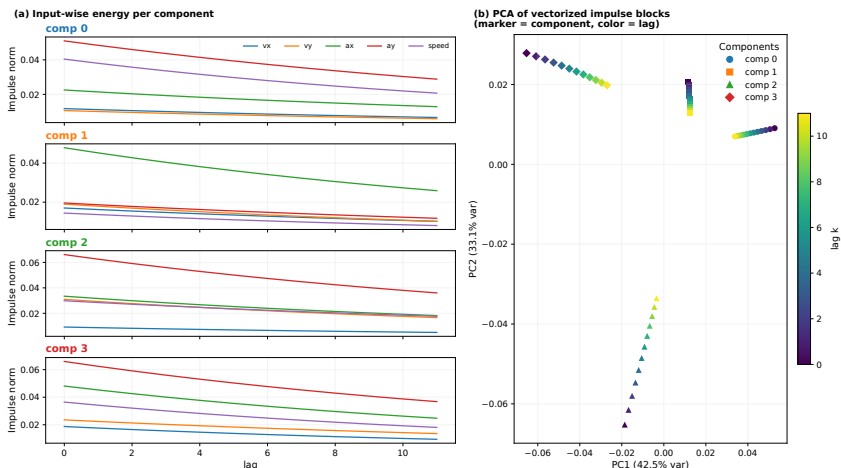

Figure 14: **Input specificity and geometry of Markov responses:** Left: input-wise energies versus lags are shown per component. Right: PCA of vectorized impulse blocks (marker shape = component, color = lags).

where $\text{speed} = \sqrt{v_x^2 + v_y^2}$ and $Y_t$ denotes PCA-reduced, $z$-scored firing rates. Here we use $p = 16$ PCs, which explain $> 90\%$ of neural activity. As in Area2 (App. E.1), angles are used only for analysis/visualization; MoLDS is trained without angle labels.

**Tensor-EM MoLDS train/val/test setup.** Training and selection mirror Area2: tensor or random initialization followed by EM refinement; Model selection was performed on the validation split. For PMd, we sweep $K \in \{2, 3, 4\}$, $n \in \{4, 5, 6\}$, and Markov horizon $L \in \{12, 16, 24\}$. The selected model in our experiments is $K=4$, $n=5$, $L=12$.

**Analysis idea.** We reuse the analysis logic from Area2. As the direction angles are not discrete as in Area2, we split them into 12 bins, and each bin is colored by the dominant component (largest mean $r_{ik}$ in the bin). Component-wise preferred directions are shown by responsibility-weighted arrows

$$v_k = \sum_i r_{ik} \left[\cos \theta_i, \sin \theta_i\right],$$

plotted as an inset (angle $\angle v_k$, length $\|v_k\|$). For each component, we examine the Markov parameter blocks

$$g_k(\tau) = C_k A_k^\tau B_k \in \mathbb{R}^{p \times m}, \qquad \tau = 0, \ldots, L - 1,$$

and plot (a) the compact magnitude curves $\|g_k(\tau)\|_F$ and (b) per-input energies $E_{k,j}(\tau) = \|g_k(\tau)_{:,j}\|_2$ for $j \in \{v_x, v_y, a_x, a_y, \text{speed}\}$. For a global view, we also vectorize $\{g_k(\tau)\}_{k,\tau}$, run PCA, and scatter the first two PCs.

Figure 14 (a) shows per-input energies versus lag, revealing input selectivity and decay rates. Figure 14 (b) visualizes the geometry of vectorized impulse blocks, where marker shape represents component identity and color denotes the lag. These demonstrate that MoLDS components exhibit separated clusters corresponding to component-specific dynamical subspaces.

