# OpenReview forum: "Learning Mixtures of Linear Dynamical Systems via Hybrid Tensor-EM Method"
_ICLR.cc/2026/Conference — ICLR 2026 Poster_

### Official Review · Reviewer_vVHY · 2025-10-31

**Soundness:** 4
**Presentation:** 4
**Contribution:** 4
**Rating:** 6
**Confidence:** 5

**Summary:**

The paper proposes a two-stage pipeline for learning mixtures of linear dynamical systems (LDSs). Stage one follows the tensor-decomposition framework of Bakshi et al. to obtain consistent initial estimates under standard conditions. Stage two refines all LDS parameters via an EM procedure that uses responsibility-weighted sufficient statistics (updating dynamics, emissions, and initial states). A principled initialization for the noise covariances Q and
R based on residual covariances is introduced. The method is applied to neural recordings from primate somatosensory cortex during reaching, illustrating the approach on a scientifically meaningful dataset.

**Strengths:**

The paper is exceptionally well written and easy to follow.

The two-stage design is natural: tensor methods provide strong initializers with theoretical guarantees, while EM addresses the well-known limitations of pure tensor approaches by locally refining parameters.

Assumptions are clearly stated and appropriate (persistent excitation; controllability/observability; component separation), with careful discussion of when they hold.

A notable contribution is the explicit treatment of Q and R, which are often sidelined in prior tensor-based work; the residual-covariance initialization is principled and practical.

The overall study exemplifies well-designed statistical methodology: theory-grounded, reproducible, and attentive to the mechanics of LDS estimation (including filtering/smoothing considerations).

The application to neural data is compelling and broadens the paper’s relevance beyond synthetic studies.

**Weaknesses:**

I can’t identify any obvious weaknesses, other than that it’s hard for me to evaluate data collected on primates beyond evaluating the statistical methodology, which as I’ve stated before is rock solid.

**Questions:**

What further applications to the authors see for mixtures of LDS’s?  What further improvements can be made to their statistical pipeline?

---

> ### Author Response · Authors · 2025-11-21
> **We sincerely thank the reviewer for the thoughtful evaluation and positive comments.**
>
> Regarding weakness:
>
> LDS-based models have become increasingly useful in neuroscience, where different behavioral conditions or latent cognitive states can induce distinct dynamics in neural activities. The MoLDS framework builds on this foundation and extends it to cases where multiple trials exhibit heterogeneous temporal dynamics. As the reviewer noted, the hybrid Tensor-EM pipeline leverages the complementary strengths of both stages: globally consistent initialization via tensor methods, followed by local refinement through EM. After validating these benefits on synthetic benchmarks, we applied the approach to two publicly available primate datasets involving reaching movements. These datasets provide clean alignment between neural activity and kinematic inputs, making them ideal for demonstrating input-output MoLDS inference and validating that the method can recover interpretable mixtures of latent dynamics in realistic settings.
>
>
> Regarding questions:
> Except for neuroscience applications, we think the MoLDS framework and Tensor-EM method will also work for a number of financial and engineering applications. For future directions of MoLDS, we envision several important directions on the statistical and technical aspects, now also included in the revised paper:
> 1. Formal efficiency guarantees for Tensor-EM in the MoLDS setting:
> In this work, we empirically demonstrate that the Tensor-EM framework effectively combines the global consistency of tensor initialization with the local refinement capability of EM for both synthetic and real data. A natural next step is to establish formal statistical efficiency guarantees for the EM stage when initialized by a tensor estimator in the MoLDS setting. Such guarantees would provide deeper theoretical foundations, similar to results known for EM in simpler mixture models. However, this analysis will be extremely technically challenging due to the temporal dependence and latent structure of LDS mixtures, but we think progress may be possible in certain simplified or idealized settings.
>
> 2. Extensions to autonomous LDS mixtures:
> Our current formulation focuses on input-output MoLDS. Extending the Tensor-EM pipeline to autonomous or partially observed input settings would substantially broaden applicability, especially in domains where inputs are missing, implicit, or only weakly structured.
> 3. Extensions to nonlinear scenarios:
> As noted by the other reviewers, another future direction will be to extend this framework to accommodate applications with high levels of nonlinearity. In the revised paper, we discuss possible strategies for this problem, such as segmenting trajectories into sessions that can be locally approximated by linear systems or enhancing the linear models with nonlinear kernels. The effort in this direction will further expand the MoLDS framework for more applications.
> We have added a short discussion in the revised paper (Conclusion), and we hope the proposed directions will strengthen and generalize the MoLDS framework for more real-world applications.

---

### Official Review · Reviewer_tLS4 · 2025-11-01

**Soundness:** 3
**Presentation:** 4
**Contribution:** 3
**Rating:** 8
**Confidence:** 3

**Summary:**

The paper proposes a tensor-based moment method for learning mixtures of linear dynamical systems (MoLDS). The authors use Simultaneous Matrix Diagonalization on the moment tensors constructed from the input-output data to recover globally consistent estimates of mixture weights and system parameters, which are then refined with the Kalman smoother. The paper applies the proposed method to both synthetic and real-world neural datasets.

**Strengths:**

* Clarity: The paper is extremely well written, with clear motivation and definition of the problem, related work, and where the proposed method stands (i.e., Contributions). It also has a clear explanation of the method with appropriate equations and algorithm pseudocode, and the Figures are well presented with descriptive captions.

* Evaluation: The method is tested extensively on both simulated and real-world neural datasets. The results are well presented, showing the advantage of the proposed method over the compared methods.

* Significance: The paper is well motivated, setting up the existing problem of the difficulty of learning MoLDS despite its usefulness. The paper tackles this problem and demonstrates its effectiveness in analyzing neural data.

**Weaknesses:**

As the authors noted, one of the limitations of the paper is that the proposed method is only evaluated on datasets that have linear dynamics. A more detailed explanation of what could happen when there’s a model mismatch, in addition to how the method could be extended to mitigate the mismatch, would be valuable.

**Questions:**

I am curious about how the proposed method will perform as we scale up the size of the dataset in terms of the observations and latent dimension. In general, I would be interested in stress testing the proposed method by varying different experimental axes.

---

> ### Author Response · Authors · 2025-11-21
> **We thank the reviewer for the clear summary and positive comments of our work.**
>
> We appreciate the reviewer highlighting both the methodological novelty and the practical value demonstrated on synthetic and real neural datasets.
>
> Weakness:
>
> We agree that model mismatch is an important consideration. Our current experiments focus on datasets whose underlying dynamics are well-approximated by linear models, which allows us to carefully assess the identifiability and recovery properties of MoLDS under controlled conditions. In the presence of nonlinear latent dynamics, there will be two important aspects for MoLDS and the Tensor-EM method:
> Bias in the moment estimates.
> The tensor moment construction is derived assuming the latent process with linear-Gaussian dynamics. If strong nonlinearities exist, they will distort the induced moment structure and then lead the initial tensor estimator to recover parameters corresponding to the linear mixture approximation rather than the true nonlinear system. This is the biggest challenge for tensor-based methods.
>
>
> Reduced effectiveness of EM refinement.
> Since EM assumes a linear LDS structure during smoothing and M-step updates, in highly nonlinear situations, it may converge to biased local optima when the latent dynamics deviate substantially from linearity.
>
> However, these limitations can be mitigated to a certain extent with extra steps:
> Locally linearized MoLDS: one can first segment complicated and long trajectories into regimes where linearization is accurate, and then apply MoLDS within each segment.
>
>
> Feature-expanded MoLDS: we envision that constructing approximate moment tensors in an appropriate nonlinear feature space could also mitigate model mismatch due to nonlinear latent dynamics. While the precise pipeline for such a feature-expanded MoLDS framework remains open right now, it is a promising avenue for extending moment-based methods beyond purely linear regimes.
> We have added a short discussion in the revised version to discuss these points (Conclusion).
>
> Question:
>
> Scaling is indeed an important and interesting axis for future work. Empirically, we observe:
> Tensor stage:
> The moment construction scales linearly in the number of samples, while the simultaneous matrix diagonalization step scales roughly cubic in the latent dimension (due to the SVD operations). In practice, for not extremely high latent dimensions, this stage remains efficient.
>
> EM refinement:
> The computational cost is dominated by Kalman smoothing, which scales linearly in sequence length and quadratically in latent dimension. For moderate latent dimensions, this remains tractable on long datasets.
> In the new version of the paper, we increased the dimension of fitting for the application on Area2 dataset from 6 to 20 dimensions (see Figure 11). Our method still works well and discovers meaningful results for this dataset analysis. We hope this new test will answer the reviewer’s question on “stress-test”. In the future, we will continue to conduct more comprehensive scaling studies, particularly since neuroscientific datasets continue to grow in size and dimensionality.

---

### Official Review · Reviewer_Lrdy · 2025-11-05

**Soundness:** 3
**Presentation:** 2
**Contribution:** 2
**Rating:** 4
**Confidence:** 3

**Summary:**

This paper presents a new fitting procedure for modeling heterogeneous time-series data based on Mixtures of Linear Dynamical Systems (MoLDS) by introducing a robust Tensor-EM algorithm. The authors propose combining tensor-based moment estimation with EM refinement and show that this method achieves more accurate and efficient results than alternative methods based on single LDSs (Linear Dynamical Systems) or switching LDSs. They also extend the application to neural data and exemplify the method on complex recordings of a reaching task and neural recordings from premotor cortex to demonstrate the applicability of their MoLDS framework to neuroscience.

**Strengths:**

1) The authors enhanced the MoLDS model to achieve improved numerical stability and robustness to noise via Simultaneous Matrix Diagonalization (SMD) compared to prior tensor methods.
2) The authors extend the tensor initialization paradigm by integrating it with a Kalman filter-smoother EM procedure.
3) The method, which uses LDSs, maintains representation interpretability.
4) The authors demonstrated their results on two synthetic experiments and two real-world neural datasets.

**Weaknesses:**

1) My biggest concern relates to the extension of MoLDS to neural data. Neural activity exhibits complex temporal evolution that changes over time, potentially even within a single trial. While I understand this approach enables interpretability by using a single LDS per trial, it cannot capture the temporally changing nature of brain dynamics, which is a major question in neuroscience. Hence, I wonder whether neuroscience is the right application to demonstrate the method. Alternatively, how could the method enable understanding of varying brain interactions over time? For example, could your fitting procedure be extended to more complex LDS models that change over time (e.g., SLDS or dLDS)?

2) It seems you treat the "per-direction single-LDS" as a gold standard. I understand that in real data we lack access to the true operators, so estimation is necessary. However, I am concerned that per-direction single-LDS oversimplifies the dynamics. Given the complexity of neural data, including trial-to-trial variability due to unobserved factors even under the same hand direction, this baseline may yield biased comparisons.

Have the authors examined trial-to-trial variability within a direction and its effect on your comparison? Specifically, what happens if you train both your method and the per-direction single-LDS on random subsets of trials within a direction, repeating this process with different subsets? Does performance remain stable?

With respect to that, I also think a missing comparison is to train a single LDS per trial, then perform clustering on these independently trained LDSs to verify whether they capture trial direction as you implicitly assume. After validating that LDSs cluster by direction (without forcing a single LDS on all trials from a direction), you could then compare this to your method

3) I think the paper could be written more clearly, as it is currently difficult to follow. The utility and importance of certain components are unclear. For example, more background information on SMD and its contribution to stability would be helpful in the related work section. Additionally, please explain more explicitly how SMD was applied in your method. I would also suggest dedicating space to explaining and defining the moments and W in Eq. 3. Finally, in the LDS paragraph (lines 54-59), there should be discussion of other LDS models that support multiple LDSs with mixtures, like dLDS [1,2].

4) The abstract is very long and difficult to follow. I suggest condensing everything from `We validate our framework on...' to the end into a single sentence, which would also save space in the main text.

5) I am not sure why the method illustration appears only in Figure 3. This illustration is helpful for understanding the paper and should appear earlier

5) Some smaller comments on writing and presentation issues that should be fixed: 1) Please pay attention to \cite vs. \citep, it seems all references appear as part of the text (no brackets), which hinders readability. 2) Hyphens are used instead of em dashes in several places. 3) In Fig. 4b, the subplots are ordered oddly (dim 4-0-1-5...).
- Some typos and formatting issues: extra bracket in line 466, double (c)(c) in line 240, missing space before reference in line 060.

[1] Mudrik, N., Chen, Y., Yezerets, E., Rozell, C. J., & Charles, A. S. (2024). Decomposed linear dynamical systems (dlds) for learning the latent components of neural dynamics. Journal of Machine Learning Research (JMLR)

[2] Chen, Y., Mudrik, N., Johnsen, K. A., Alagapan, S., Charles, A. S., & Rozell, C. (2024). Probabilistic decomposed linear dynamical systems for robust discovery of latent neural dynamics. Advances in Neural Information Processing Systems (NeurIPS)

**Questions:**

1) Could you clarify how you performed the SLDS comparison? Specifically, did you take the most active operator from SLDS trained on all trials together? How did you define the number of operators? What happens if you include stronger penalization on the number or frequency of switches per trial? I would assume that with strong penalization you may be able to capture the individual dominant operator as in MoLDS.

2) Regarding my first concern about non-stationarity within a trial, I am curious what happens if you split each trial into multiple periods (e.g., stimulus presentation vs. reaching period) and run your method treating these as separate trials. Do you get the same operator being active in both periods, or do they switch?

3) Have you examined multi-step reconstruction instead of single-step?

4) It seems you focused mainly on the dominant component from MoLDS. I am curious about the distribution of probabilities for different LDSs within a single trial. Is the probability of the second component, for example, very close to that of the first? If so, what does the second component look like? Is it also shared between trials of similar direction? (I.e., for a fixed and large K, unlike Figure 4b where you appear to modulate K).

5) Could you explain the rationale for choosing a test-val-train split over cross-validation?

6) Could you clarify the difference between Sections 4.1 and 4.2 in terms of which version of your method was used? For instance, in Section 4.1, did you use only SMD with randomly initialized EM?

---

> ### Author Response · Authors · 2025-11-21
> **We sincerely thank the reviewer for the careful evaluation of our paper and for the thoughtful comments.**
>
> W1: We would like to clarify one important point: MoLDS is designed for collections of short trajectories and assumes each trial is (approximately) generated by a single LDS. In our neural applications, we explicitly select short, movement-related epochs that satisfy this assumption (−100 to +500 ms for Area2, same for PMd). We now clearly state these assumptions in the main text. We also conduct new single-trial and per-direction LDS fits, which confirm that these brief epochs are well modeled by a single LDS (see Figs. 5, 11). Extending MoLDS to capture time-varying dynamics would require advanced tensor methods, and such extensions are non-trivial.
>
> W2: We examined trial-to-trial variability by fitting an LDS to every single trial and comparing Markov parameter vectors. As shown in Fig. 8, single-trial LDSs form compact within-direction clusters, with the per-direction LDS near each cluster center, while a global LDS lies far from all clusters. Clustering these independently fitted single-trial LDSs also recovers movement directions (Fig. 5), confirming that direction-specific dynamics are already present at the single-trial level. These analyses show that while the per-direction LDS is not a perfect “gold standard”, but a reasonable and stable reference, and our comparisons to MoLDS capture meaningful structure in the data. We include these analyses and an additional baseline in the revision (see Sections 5.1, E.1).
>
> W3: In the revision, we made several substantial updates, including to the writing. We expand the motivation for using MoLDS in neural data analysis. And, following the reviewer’s suggestion, we incorporate a dedicated paragraph discussing LDS-based approaches, including SLDS and dLDS, to better differentiate MoLDS.
>
> W4: We have shortened the abstract as suggested.
>
> W5: Now, the figure of the overview of MoLDS is moved to Section 3.
>
> W6: We have carefully revised the paper and corrected the typos.
>
> Questions
> 1. We trained an S-state SLDS on all trials jointly, using the same Area2 data. Each SLDS state corresponds to a global linear operator. After training, we inferred the posterior state sequence for each trial and defined its dominant operator as the state with the highest occupancy time.  We set the number of SLDS operators K to match the MoLDS model’s number of components selected on the validation set, i.e., 3 for Area2. We also checked nearby values of K and obtained qualitatively similar conclusions: the SLDS states tended to represent global “phases” shared across many trials, rather than assigning a distinct dominant operator to each trial as MoLDS does. We used a mild stickiness prior, and in additional runs, we increased the stickiness to penalize switching more strongly. With very strong penalization, the model collapses toward a single global LDS. Therefore, we did not find that the SLDS produces “one operator per trial” in the MoLDS sense. We have added the results of the different stickiness of SLDS to the Appendix, in Fig 9. This experience also highlights one of the main weaknesses of SLDS - its inability to explicitly use a trial structure which is prevalent in neuroscience experiments.
> Overall, SLDS is a powerful model if one aims to capture switches in unstructured data, but MoLDS is better at capturing between-trial/condition mixture structure.
>
>
> 2. This concern touches on an important distinction between MoLDS and SLDS, which we have now clarified in the revision. MoLDS assumes each trajectory is generated by one LDS and requires trajectories to have equal length due to the lag-stacking used in tensor methods. Thus, applying MoLDS separately to stimulus/delay/movement epochs of differing lengths is not currently feasible. For Area2 and PMD data analysis, we use movement-related segments, which are approximately stationary.
>
> 3. We qualitatively evaluated multi-step predictions. An example is included in the Appendix (see Fig. 12).
>
> 4. We analyzed the full responsibility distribution to assess whether non-dominant components carry substantial probability. Across all trials, the mean dominant responsibility is 0.976, and the mean second-largest is 0.023 (see Fig. 13 for details).
>
> 5. MoLDS is trained on many independent trials, not one long sequence. Therefore, we do not split individual trajectories into folds like SLDS for cross-validation. Because real data lacks ground-truth mixture structure, we require model selection (for K, n), which we perform by using a validation set with metrics BIC/NLL/RMSE. At last, we use the test set for the final assessment of clustering and prediction quality.
>
> 6. Section 4.1 is the comparison of the pure tensor method (SMD and RTPM) on simple synthetic data. In Section 4.2, we then evaluate the Tensor-EM method on higher-dimensional synthetic data where SMD is implemented for tensor initialization.

---

### Official Review · Reviewer_jFVZ · 2025-11-09

**Soundness:** 3
**Presentation:** 2
**Contribution:** 2
**Rating:** 2
**Confidence:** 3

**Summary:**

This paper develops a new approach towards fitting Mixture of Linear Dynamical Systems models, using tensor-based methods followed by EM refinement. The paper demonstrates the superiority of their model-fitting approach on simulated data. The paper then fits the model to two neural activity datasets.

**Strengths:**

-The authors demonstrate a model that (to my knowledge) hasn't been used in neuroscience data before
-The authors show convincing results of the benefit of their new tensor-EM fitting approach in simulated data (and quantitatively versus standard EM approaches on one real neural dataset)

**Weaknesses:**

The importance of MoLDS for modeling neural data (which is the focus of this paper) never got across clearly to me. It feels like the authors take that importance as a given, but this is a method that hasn't been used (to my knowledge) in neural data, so really clearly motivating the need is essential - at the moment, it's primarily a sentence on line 60-61. When are specific examples when a neuroscience researcher would want to use MoLDS when existing approaches would fail? The real data examples don't make this clear, since it seems like you could just fit LDS models to different reach directions in those scenarios (they don't show any benefit of an unsupervised approach), and it's especially unclear what to take away from the results on the PMd dataset.

Additionally, the comparisons to alternative approaches for the real datasets were lacking. There was not a comparison to the tensor approach to fitting, so it's currently not clear whether this paper's fitting approach is advantageous relative to pure tensor methods. Additionally, while the paper shows a numerical advantage versus a pure EM approach, there's no advantage shown in terms of the model analysis (e.g. within Fig 4c). So in general, the advantages of this paper's model-fitting approach aren't currently shown strongly.

**Questions:**

1. It took me a very long time into the paper to really understand how MoLDS was different than an SLDS. It would help readers a lot if you more explicitly compared them early on in the methodology, since SLDS is much more common in comp neuro (which will be the primary readers of this paper).
2. I'd clarify why the RTPM method is being used for a comparison - what past paper was this used in?
3. It seems very odd to be doing PCA on Area 2 data prior to fitting MoLDS, when generally an advantage of LDS models is that they are directly learning the underlying latent states from high-D data.
4. Relatedly, does the PMd experiment also use PCs or all neurons?
5. Line 72-79 - if possible, some citations for pros/cons of approaches could be useful (although I realize sometimes these insights are from personal experience)

---

> ### Author Response · Authors · 2025-11-21
> **We sincerely thank the reviewer for these thoughtful comments.  Below, we provide detailed responses to the noted weaknesses and questions.**
>
> Weaknesses:
> Neural experiments often contain a trial structure, with one parameter of interest that varies across trials, such as the direction of reach in a motor setting. This change in stimulus and the resulting behavior may lead to a change in neural dynamics. This structure naturally lends itself to analysis of dynamics that may differ across trials in vastly different conditions, or may be similar across trials in conditions that are more alike. This is the setting in which our MoLDS model is very helpful: towards identifying clusters of neural dynamics across trials or across conditions. This is different from the popular SLDS model in that it actively uses the trial structure that is an experimental constraint. The SLDS model, while a very appropriate model for understanding underlying neural dynamics in unstructured setting, may not help in understanding across-trial structure (see Fig 9 for SLDS on Aear2). On the other hand, researchers also use a separate dynamical model per condition or per trial, and then try to understand the relationship between the different learnt models.
>
> Here, instead of fitting a model to each trial or each condition separately and then calculating the similarity in parameters to assess similarity across conditions, MoLDS can provide a potential way to evaluate all trials simultaneously and let us investigate the mixture structures in the entire dataset and the underlying LDS components. In the original version of the paper, we did not clearly state the motivation due to the page limit, which led to some confusion. Now, we have edited the corresponding paragraph to explicitly clarify this motivation.
>
> The benefit of this model is shown in the area2 dataset, where we have clear direction-related conditions that can be used to separate the trials, a structure that SLDS fails to recover. Our MoLDS result, on the other hand, is able to separate out trials according to condition similarity, and is consistent with the supervised single-direction fitting and also similar to the newly added single-trial LDS results. The benefit is further pronounced in the PMd dataset, where the task is a sequential movement, and there are no clear directions or condition structure to separate each trial; MoLDS can still provide a reasonable characterization of the trials and underlying dynamics.
>
> In addition to the original comparisons, we now add the comparison of the pure tensor method results for the Area2 and PMd applications in Figure 5. Moreover, we add the recovered states using Random-EM as well.
>
>
> Q1: SLDS is designed for long, nonstationary sequences where the dynamics themselves evolve over time. In contrast, the MoLDS setting assumes that each short trajectory is well described by a single LDS drawn from a collection of LDS components. The goal is to identify the set of LDSs and assign trials to those LDS components, rather than modeling dynamic changes within unstructured data. In many neuroscience applications, trial structures are still used very heavily, and this model is more appropriate for these situations. In theory, an SLDS model may recover the different trials as different LDS’s, but in practice this does not seem to always take place (see Figure 9 where the different trials are not cleanly parsed into different states in an SLDS), making it difficult to understand trial-level differences in the data.
> We now modify the text to move the schematic of MoLDS to the Methods section, which would make it easier for readers to quickly understand the formulation and setup.
>
> Q2: The RTPM method was used in a recent paper (Rui&Dahleh,2025) of MoLDS for the tensor decomposition step. Here, we adopt the SMD method and empirically show the improved performance for the tensor-based parameter estimations by comparing with RTPM. The citation was previously given elsewhere in the paper, and we now give the citation again at place.
>
> Q3: Here, we apply PCA as a mild denoising step to reduce noise for fitting MoLDS. The main reason is that several substeps in the tensor initialization, such as moment estimation, require well-conditioned matrices to obtain reliable initial estimates for EM refinement. With very high-dimensional observations, these steps become numerically unstable, which is a known challenge for moment-based identification methods on large-scale datasets. Importantly, we verified that the first 6 principal components explain around 90% of the total variance in the Area2 dataset (see Figure 7). Moreover, in new simulations in the revised paper, we have increased the PC dimension to 20 and then applied our method (see Figure 11). The MoLDS results match the clustering patterns obtained from per-direction and single-trial LDS fits, as in the original version.
> Q4: Similarly to above, we also use 16 PCs for the PMd dataset, which takes up >90% of the total variance in this dataset.
> Q5. We have added suitable citations now.

---

> > ### Comment · Reviewer_jFVZ · 2025-11-27
> >
> > Thanks for your thoughtful response, but I'm still not convinced about the usefulness of the MoLDS approach for neural data analysis, which is the paper's primary motivation. What scientific hypotheses can be developed based on the PMd results, which is an example where you claim that MoLDS would be beneficial? Or what is another specific example of an experimental paradigm where MoLDS would show the most promise?

---

> > > ### Author Response · Authors · 2025-11-27
> > >
> > > Thank you for your willingness to engage with our work. Our goal is to identify the underlying neural dynamics in trial-structured experiments, where each trial typically consists of a cue, stimulus, and behavioral response. MoLDS rests on two assumptions:
> > >
> > > (a) trial-level population activity is well described by low-dimensional linear dynamics, and
> > >
> > > (b) only a finite set of such dynamical regimes is needed to explain the data.
> > >
> > > Both assumptions are strongly supported in systems neuroscience.
> > >
> > >
> > > (a) Low-dimensional linear dynamics are widely justified: A large body of work has shown that population activity in many brain areas evolves within low-dimensional manifolds, and that linear approximations provide accurate local descriptions of these dynamics [1-3].
> > >
> > > (b) A finite repertoire of dynamical regimes is biologically motivated: Prominent theories of motor control posit that different movements are generated by reusing a limited set of dynamical motifs within the same neural circuit [4-6]. Neuroscience experiments routinely manipulate stimuli or contexts in a systematic way across many repeated trials; MoLDS formalizes the hypothesis that these conditions may recruit a shared, finite set of underlying dynamical systems. When the stimulus varies continuously, as in the PMd dataset, MoLDS provides a principled way to test whether neural activity nonetheless organizes into a discrete set of dynamical modes.
> > >
> > > Specifically in the PMd data, we find that reaches between ~270° and 30° share a common dynamical regime despite differing behavioral trajectories. This suggests that PMd may group movements with similar preparatory demands or control policies into the same underlying dynamical mode. This leads to testable hypotheses such as:
> > >
> > > -- PMd may partition the continuous space of movement directions into a small number of dynamical “families”.
> > >
> > > -- These shared dynamics may reflect common upstream planning computations or recurrent circuit motifs.
> > >
> > > -- Downstream regions (e.g., M1, spinal circuits) may differentiate these movements through separate transformations, even when premotor dynamics are similar.
> > >
> > > Such findings provide a mechanistic handle on how premotor cortex organizes behaviorally diverse movements into a small set of reusable dynamical primitives.
> > >
> > > Broadly speaking, MoLDS is well-suited to any task where multiple conditions may share latent dynamics but differ in observable trajectories. Examples include sensory decision-making with different stimulus strengths, where evidence accumulation circuits may reuse a discrete set of integration dynamics across conditions; context-switching tasks in prefrontal cortex, where the same population may cycle among a handful of rule-dependent dynamical modes; motor learning experiments, where early, mid, and late learning stages could be captured as transitions among a small set of dynamical regimes.
> > >
> > > MoLDS also offers concrete practical benefits:
> > >
> > > -- Instead of fitting a full LDS per trial, which is statistically impractical for short neural trajectories, MoLDS estimates a small number of shared LDS parameters, greatly improving identifiability.
> > >
> > > -- With a finite set of dynamical regimes, the dynamics underlying a new trial can be inferred rapidly, enabling faster and more reliable forward prediction of neural activity. This is essential for real-time neural perturbation or brain-machine interface applications. Our 1-step prediction results (Figs. 5b, 6b) show this capability.
> > >
> > > The main focus of this paper is the *inference* of MoLDS, for which we devise a new approach based on tensor methods combined with EM updates. This has utility towards neuroscience data, as concretely shown in the submission with the two datasets. Moreover, MoLDS can be widely applied towards diverse applications such as for recovering a variety of clinically-relevant behavioral dynamics [7] or identifying cell-type specific dynamics [8].
> > >
> > > [References to follow]

---

> > > > ### Author Response · Authors · 2025-11-27
> > > >
> > > > References:
> > > >
> > > > [1] Saxena, Shreya, and John P. Cunningham. "Towards the neural population doctrine." Current opinion in neurobiology 55 (2019): 103-111.
> > > >
> > > > [2] Sussillo, David, and Omri Barak. "Opening the black box: low-dimensional dynamics in high-dimensional recurrent neural networks." Neural computation 25.3 (2013): 626-649.
> > > >
> > > > [3] Brunton, Bingni W., et al. "Extracting spatial–temporal coherent patterns in large-scale neural recordings using dynamic mode decomposition." Journal of neuroscience methods 258 (2016): 1-15.
> > > >
> > > > [4] Shenoy, Krishna V., Maneesh Sahani, and Mark M. Churchland. "Cortical control of arm movements: a dynamical systems perspective." Annual review of neuroscience 36.1 (2013): 337-359.
> > > >
> > > > [5] Rokni, Uri, and Haim Sompolinsky. "How the brain generates movement." Neural computation 24.2 (2012): 289-331.
> > > >
> > > > [6] Churchland, Mark M., and John P. Cunningham. "A dynamical basis set for generating reaches." Cold Spring Harbor Symposia on Quantitative Biology. Vol. 79. Cold Spring Harbor Laboratory Press, 2014.
> > > >
> > > > [7] Bulteel, Kirsten, et al. "Clustering vector autoregressive models: Capturing qualitative differences in within-person dynamics." Frontiers in Psychology 7 (2016): 1540.
> > > >
> > > > [8] Luecke, Stefanie, Katherine M. Sheu, and Alexander Hoffmann. "Stimulus-specific responses in innate immunity: Multilayered regulatory circuits." Immunity 54.9 (2021): 1915-1932.

---

### Comment · Area_Chair_5cs3 · 2025-11-26
**Reminder to Engage!**

Dear Reviewers,

We are one week away from the end of the discussion period and the review responses have been posted. If you have not done so already, please read the response and check if the authors have addressed your concerns. Also please acknowledge the review by responding and stating how the response (and updated manuscript if provided) does or does not change your evaluation of the work. Earlier responses allow for meaningful engagement and potential for further clarification.

-Area Chair

---

### Meta-Review · Area_Chair_BS3z · 2026-01-08

**Summary:**

The paper provides an efficient algorithm for estimating the parameters of a mixture of linear dynamical systems, which has neither the limitations of EM nor conventional spectral methods, with applications to neural data analysis. Some reviewers raised concerns about the appropriateness of the method for neural data, while others praised the clarity of the presentation. While there are legitimate questions about the value of said model for generating useful scientific hypotheses in neuroscience, for ICLR an emphasis primarily on methods development is fine. I recommend acceptance.

**Reviewer Concerns:**

Reviewer jFVZ specifically held out that the method did not seem to clearly provide a justification for use in neural data analysis, as they could not see how the model could be used to generate interesting scientific hypotheses. However I believe this is less of a concern for ICLR (a neuroscience journal would be a different matter).

**Reviewer Scores:**

I believe reviewer Lrdy might have bumped their score up if able to engage. I'm also surprised that reviewer vVHY gave the paper as low a score as they did given the content of the review.

---

### Decision · Program_Chairs · 2026-01-26

Accept (Poster)